# Odd viscosity in chiral active fluids

Debarghya Banerjee[1], Anton Souslov [1,2], Alexander G. Abanov[3] & Vincenzo Vitelli [1,2]

We study the hydrodynamics of fluids composed of self-spinning objects such as chiral grains or colloidal particles subject to torques. These chiral active fluids break both parity and time-reversal symmetries in their non-equilibrium steady states. As a result, the constitutive relations of chiral active media display a dissipationless linear-response coefficient called odd (or equivalently, Hall) viscosity. This odd viscosity does not lead to energy dissipation, but gives rise to a flow perpendicular to applied pressure. We show how odd viscosity arises from non-linear equations of hydrodynamics with rotational degrees of freedom, once linearized around a non-equilibrium steady state characterized by large spinning speeds. Next, we explore odd viscosity in compressible fluids and suggest how our findings can be tested in the context of shock propagation experiments. Finally, we show how odd viscosity in weakly compressible chiral active fluids can lead to density and pressure excess within vortex cores.

[1] Instituut-Lorentz, Universiteit Leiden, Niels Bohrweg 2, Leiden 2333 CA, The Netherlands. [2] The James Franck Institute and Department of Physics, The University of Chicago, Chicago, IL 60637, USA. [3] Department of Physics and Astronomy and Simons Center for Geometry and Physics, Stony Brook University, Stony Brook, NY 11794, USA. Debarghya Banerjee and Anton Souslov contributed equally to this work. Correspondence and requests for materials should be addressed to V.V. (email: vitelli@uchicago.edu)

Chiral active fluids are materials composed of self-spinning rotors that continuously inject energy and angular momentum at the microscale. Out-of-equilibrium fluids with active-rotor constituents have been experimentally realized using nanoscale biomolecular motors[1–7], microscale active colloids[8–10], or macroscale-driven chiral grains[11]. In order to unlock the potential of chiral active matter for the design of materials with novel functionalities, one needs to understand the excitations and the mechanical response to perturbations around their far-from-equilibrium steady states. In some cases, time-reversal-symmetry breaking endows these systems with geometrical and topological features reminiscent of quantum Hall fluids and topological insulators[12,13]. For example, chiral active media can support topologically protected excitations such as chiral edge modes, which are responsible for the unidirectional propagation of density waves[14]. We proceed by discussing the mechanical response of far-from-equilibrium steady states.

The mechanical response of any viscoelastic material is typically encoded in its constitutive relations: a set of equations that express the stress tensor in terms of the strain and strain rate[15]. Conservation of angular momentum dictates that the stress tensor $\sigma_{ij}$ of any medium with vanishing bulk external torque must be symmetric under the exchange of its two indices $i$ and $j$. This conclusion, however, does not apply to chiral active fluids composed of self-spinning constituents (Fig. 1a) that are driven by active torques[16–21]. In addition to the presence of an antisymmetric stress[11,21–25], chiral active media exhibit anomalies in the symmetric component of $\sigma_{ij}$ that encodes the viscous stress.

In this paper, we ask a deceptively simple question: what is the viscosity of a chiral active fluid? Viscosity typically measures the resistance of a fluid to velocity gradients. It is expressed mathematically by a tensor, $\eta_{ijkl}$, that acts as a coefficient of proportionality between viscous stress $\sigma_{ij}$ and strain rate $v_{kl}$[15]. The Onsager reciprocity relation stipulates that $\eta_{ijkl}$, like any such linear transport coefficient, must be symmetric (or even) under the exchange of the first and last pairs of indices (i.e., $\eta_{ijkl} = \eta_{klij}$) provided that time-reversal symmetry holds[15]. Here we show how such chiral active fluids break both parity and time-reversal symmetries in their steady states, giving rise to a dissipationless linear-response coefficient called odd (or Hall) viscosity $\eta_{ijkl}^o(= -\eta_{klij}^o)$ in their constitutive relations. Avron et al. first recognized that a two-dimensional electron fluid can display a Hall viscosity in the presence of an external magnetic field that breaks time-reversal symmetry (TRS) at equilibrium[26–32]. In chiral active fluids, violation of Onsager reciprocity originates from the breaking of microscopic reversibility out of equilibrium, a feature inherent to active matter[33,34]. In this case, an odd viscosity can emerge as a linear-response coefficient calculated around the non-equilibrium steady state of a purely classical system. Despite its universal nature, odd viscosity was neglected in previous hydrodynamic theories of active rotors[11,21–25] that implicitly consider only rotors with small spinning frequency near equilibrium—a regime for which the antisymmetric stress dominates over the odd viscosity. On general grounds[28,29], it can be shown that odd viscosity is proportional to the non-vanishing net angular momentum density that exists within the active fluid in steady state.

## Results

**Hydrodynamics of chiral active rotors.** For concreteness, we construct a hydrodynamic description of "dry" chiral active fluids based on a constitutive relation that explicitly accounts for an odd viscosity term. Here the term "active matter" designates those

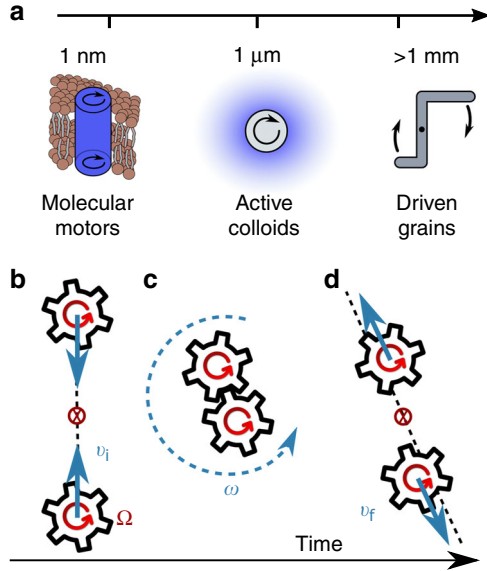

**Fig. 1** Chiral active fluids with odd viscosity. **a** Chiral active fluids in a variety of contexts: biological[1–6], colloidal[8–10], and granular[11]. **b–d** Schematic of the collision processes in a chiral active gas. **b** Head-on collision between self-spinning gears that initially move with speed $v_i$ and rotate with frequency $\Omega$. Their center of mass is represented as a red crossed circle. **c** While in contact, the frictional gears convert intrinsic angular momentum into orbital angular momentum, which leads to rotation around their center of mass with frequency $\omega$ (shown as a dashed circular arrow). We assume that this process occurs on a time scale that is fast compared to the time between collisions. As a result, the spinning frequency is rapidly reset to the initial $\Omega_0$ favored by the balance of internal active torque and dissipation. **d** After the collision, the self-spinning gears move away from each other with velocity $v_f$. However, the particles do not necessarily move with final velocities that parallel the vector distance between them, i.e., orbital angular momentum is generated in the collision

fluids in which kinetic energy is not conserved. In such materials, momentum, including angular momentum, does not have to be conserved. In dry active matter, momentum is dissipated via friction (e.g., with substrate), although this effect could be small relative to inter-particle interactions.

In two dimensions, the evolution of the slow variables, i.e., density of particles $\rho$, center-of-mass velocity components $v_i$, and intrinsic spinning frequency of the rotors $\Omega(\mathbf{x}, t)$, is governed by the following equations (see Methods section for detailed derivations):

$$D_t\rho = -\rho\nabla\cdot\mathbf{v}, \tag{1}$$

$$\rho D_t v_i = \partial_j\sigma_{ij} - \Gamma^v v_i. \tag{2}$$

$$I D_t\Omega = \tau + D^\Omega\nabla^2\Omega - \Gamma^\Omega\Omega - \epsilon_{ij}\sigma_{ij}, \tag{3}$$

where $D_t(\equiv\partial_t + v_k\partial_k)$ denotes a convective derivative and $I$ is the moment-of-inertia density. A constant active torque applied to each rotor and described by the torque density $\tau$ injects energy into the fluid, thus breaking detailed balance.

The hydrodynamic Eqs. (1)–(3) originate from the forces and torques experienced by particles with internal active rotation. Equations (1) and (2) are simply the equations of motion for any fluid with the addition of substrate friction—expressing the conservation of mass via the continuity Eq. (1) and the evolution of fluid momentum, Eq. (2). The additional equation peculiar to

the chiral active fluid is Eq. (3) that governs the evolution of the intrinsic spinning frequency $\Omega(\mathbf{x}, t)$. Equations of this form have been considered in previous hydrodynamic theories of chiral active fluids in various regimes, see e.g., refs. [11,21–25]. Activity enters the system via the torque $\tau$ in Eq. (3). In addition to the active torque, Eq. (3) includes the dissipation of intrinsic angular momentum via $\Gamma^\Omega$: this term is responsible for the rapid equilibration of the system if the active torque were to be turned off. The rotational dissipation coefficient $\Gamma^\Omega$ saturates energy injection in a way that leads to a non-equilibrium steady state with non-vanishing single-rotor frequency $\Omega \sim \tau/\Gamma^\Omega (\equiv \Omega_0)$, whereas $D^\Omega$ and $\Gamma^v$ control diffusion of intrinsic rotations and linear momentum damping, respectively. The left-hand side of Eq. (3) includes the convection of intrinsic rotation due to the center-of-mass motion of the fluid $v_i$, whereas the right-hand side of Eq. (3) accounts for the coupling between the intrinsic and orbital angular momentum via the fluid stress $\sigma_{ij}$.

The stress in Eqs. (2) and (3) is given by

$$\sigma_{ij} \equiv \epsilon_{ij} \frac{\Gamma}{2}(\Omega - \omega) - p\delta_{ij} + \eta_{ijkl}v_{kl} + \frac{\ell}{2}\left(\partial_i v_j^* + \partial_i^* v_j\right), \quad (4)$$

where $\omega \equiv \frac{1}{2}\epsilon_{ij}\partial_i v_j$ is the vorticity, $v_j^* \equiv \epsilon_{jl}v_l$ is the velocity vector rotated clockwise by $\pi/2$, and $\ell \equiv I\Omega$ is the intrinsic angular momentum density. (Note that $\epsilon_{ij}$ denotes the Levi–Civita antisymmetric tensor in 2D.) The stress is composed of the usual fluid stress terms due to the pressure $p$ and the (dissipative) viscosity tensor $\eta_{ijkl}$ present in any fluid, and two terms peculiar to chiral active fluids. One such term is the antisymmetric stress in Eq. (4) proportional to $\Gamma$, which results from inter-rotor friction and couples the flow $\mathbf{v}$ to the intrinsic rotations $\Omega$[11,21–25].

The last component of the stress $\sigma_{ij}$ is the novel ingredient that we add to Eq. (4), which we derive in the Methods section using a hydrodynamic variational principle. To arrive at such a term, we must include a coupling between vorticity and intrinsic angular momentum. To do so, we start with an action that includes both the (standard) action necessary to arrive at the Navier–Stokes equations and an additional term $\int \omega \ell d^2x dt$, which also appears, e.g., in the variational hydrodynamics of magnetofluids[35]. Notably, the variational approach that we take to arrive at the last term in Eq. (4) does not account for energy dissipation. Unlike the dissipative viscosity $\eta_{ijkl}$, this extra nonlinear term is derived based on energy conservation, even in the presence of other active and dissipative terms. Our derivation of Eq. (4) proceeds by taking an action with Clebsch parameters and varying it with respect to all of the dynamical fields[36,37]. The Clebsch parameters are auxiliary variables that allow us to write a Lagrangian density from which we can derive the fluid equations using Hamilton's equations of motion. The resulting equations can be reduced to just the ones in Eqs. (2) and (3) by substituting in the Clebsch parameters for hydrodynamic variables, and at the end of this substitution, the term $\frac{\ell}{2}\left(\partial_i v_j^* + \partial_i^* v_j\right)$ emerges as part of the fluid stress tensor (see Methods). This term is a nonlinear coupling between the fields $\mathbf{v}(\mathbf{x}, t)$ and $\ell(\mathbf{x}, t)$ [or, equivalently, $\Omega(\mathbf{x}, t)$] that was neglected in previous linear hydrodynamic theories of active matter[11,21–25].

We linearize the equations around a constant $\ell$ characteristic of the non-equilibrium steady state, in which parity is broken: in this state, the nonlinear coupling reduces to an anomalous transport coefficient $\eta^o = \ell/2$ called odd viscosity in the constitutive relations, Eq. (4). Note that if the equations are linearized around $\ell = 0$, as appropriate for the relaxation dynamics of passive, undriven fluids, the odd viscosity term disappears. On the other hand, if the active torque dominates over the inter-rotor coupling $\Gamma$, the ensemble of self-spinning rotors behaves as a weakly

interacting chiral active gas with a constant angular momentum $\ell = I\Omega_0$. In such a chiral gas, the active rotation frequency is near $\Omega_0$ for each particle except during and immediately after each collision, when some intrinsic rotation is converted into fluid vorticity by the antisymmetric stress (Fig. 1b). While this conversion is crucial in establishing the chiral steady state of a gas of rotors, the state itself depends only on odd viscosity $\eta^o$ and not on the inter-rotor coupling, if $\Gamma$ is sufficiently small.

**Constant intrinsic rotation rate**. We now determine the conditions for the emergence of odd viscosity. Gradients of intrinsic angular rotation $\Omega$ are negligibly small if the characteristic velocity $v_0$ and length scale $r_0$ are such that $v_0/r_0$ is much greater than $\Gamma/I$ and, simultaneously, much less than $\tau/\Gamma$. For this to hold, a necessary condition is that $\tau$ be much greater than $\Gamma^2/I$. In this regime, Eq. (2) decouples from Eq. (3). Furthermore, whereas the damping $\Gamma^v$ might be significant in many realizations of active-rotor systems, we neglect it for simplicity in what follows. For the effects of odd viscosity to dominate over translational friction, the damping coefficient must satisfy $\Gamma^v/I$ is much less than $\tau/(r_0^2\Gamma^\Omega)$, consistent with the large-torque regime considered here. Under these conditions, Eq. (2) reduces to the modified Navier–Stokes equation:

$$D_t v_i = \nu \nabla^2 v_i + \nu^o \nabla^2 \epsilon_{ij}v_j - \frac{\partial_i p}{\rho}, \quad (5)$$

with a familiar kinematic viscosity $\nu (\equiv \eta/\rho)$ and an additional odd viscosity $\nu^o (\equiv \eta^o/\rho)$ term. We have derived this equation in the context of a weakly interacting chiral active gas; it was written based on symmetry considerations in ref. [27]. We emphasize that the presence of the odd viscosity term results from the breaking of parity and time-reversal symmetries. Therefore, the phenomenological consequences of odd viscosity that we now discuss can apply to a larger class of chiral fluids. We now proceed to analyze Eq. (5) within this novel context and focus on those active-fluid phenomena which odd viscosity may influence and suggest possible experimental tests. The field $\Omega(\mathbf{x}, t)$ has been integrated out from Eq. (5): the only vestige of its presence is the emergent transport coefficient $\nu^o$. Leading-order corrections to Eq. (5) in gradients of $\Omega$ are captured by the antisymmetric stress, the first term in Eq. (4). The effective theory embodied by Eq. (5) ceases to be valid whenever large spatial gradients of $\Omega(\mathbf{x}, t)$ are created by interactions between rotors (e.g., at large densities)—in that case we resort to the full Eqs. (1) and (2).

Inspection of Eq. (5) reveals that the odd viscosity term is a transverse linear-response coefficient describing forces $f_i$ due to gradients in the perpendicular flow components $\epsilon_{ij}v_j$. In addition, $\nu^o$ is odd under either parity $P$ or time-reversal $T$ symmetries: $P\nu^o = T\nu^o = -\nu^o$ and, thus, it is nonzero only if both $P$ and $T$ are broken. These conclusions are consistent with the Onsager relation generalized to the case of broken $T$-symmetry, which reads $T\eta_{ijkl} = \eta_{klij}$. For an isotropic fluid, the parity-odd nature of $\eta^o$ follows from the explicit form of the viscosity tensor,

$$\eta_{ijkl}^o = \frac{1}{2}\eta^o\left(\epsilon_{ik}\delta_{jl} + \epsilon_{il}\delta_{jk} + \epsilon_{jk}\delta_{il} + \epsilon_{jl}\delta_{ik}\right), \quad (6)$$

so that $T\eta_{ijkl}^o(\eta^o) = \eta_{klij}^o(-\eta^o)$. The symmetry of the viscous component of the stress tensor ($\sigma_{ij}$) guarantees that $\eta_{ijkl}^o$ is $P$-even. This symmetry implies that $\eta^o$ and $\nu^o$ are $P$-odd, because every term in Eq. (6) also contains one $P$-odd Levi–Civita tensor. The main result is that the odd viscosity term $\nu^o\nabla^2\epsilon_{ij}v_j$ is $T$-invariant and thereby reactive: unlike dissipative viscosity $\nu$, odd viscosity

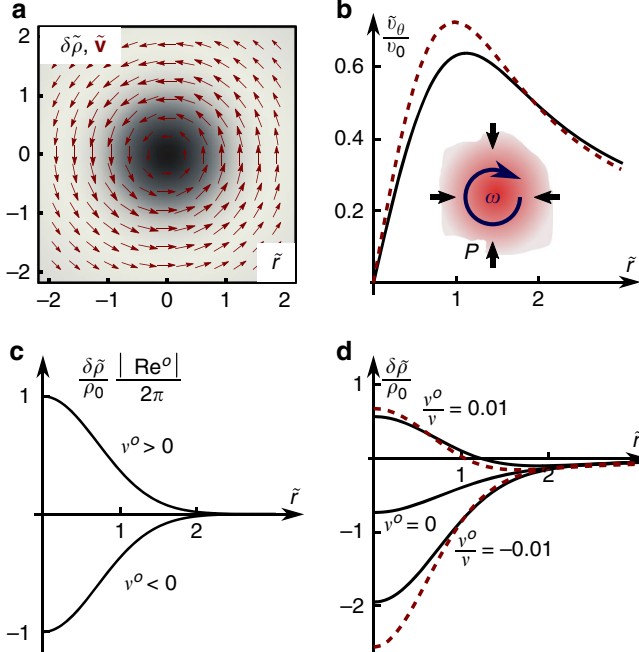

**Fig. 2** Odd viscosity in weakly compressible active fluids can lead to a build-up of particles inside a vortex. The Lamb–Oseen vortex flow (**a**: red arrow, **b**: Rescaled azimuthal component $\tilde{v}_\theta \equiv v_\theta r_s \pi / r_0$, $r_s \equiv r_0 \sqrt{1 + t/t_0}$, $r_0$ is a length scale that we chose via the initial vortex profile, and $\tilde{r} \equiv r/r_s$) is unperturbed by the presence of odd viscosity $\nu^o$ (**b**: black, solid), but does change due to antisymmetric stress (**b**: red, dashed). (**b** inset: schematic illustration of the fact that a fluid with odd viscosity couples vorticity and pressure, i.e., the phenomenon illustrated quantitatively in parts **a**–**d**.) Odd viscosity does change the density profile (grayscale map of **a**; **c**, **d**). We plot time-rescaled solutions for odd-viscosity vortices of particle density $\delta\tilde{\rho}/\rho_0 \equiv (\rho - \rho_0)\frac{\pi^2(1+t/t_0)}{\rho_0 \mathrm{Ma}^2}$, as a function of $\tilde{r}$. **c** Deep in the limit $\mathrm{Re}^o$ much less than 1, the inertial contribution to the density profile inside a Lamb–Oseen vortex can be neglected and the profile depends on the odd viscosity only. Plotted is the rescaled form of the density profile, also written in Eq. (7). The sign of the relative density at the vortex center depends on the relative sign of fluid vorticity and odd viscosity. **d** When the effects of inertia are non-negligible, they modify the density profile: inertia always favors a smaller density at the vortex core. Plotted in black are the exact solutions for Reynolds number $\frac{v_0 r_0}{\nu} = 0.05$ and the viscosity ratio $\frac{\nu^o}{\nu} = \{-0.01, 0, 0.01\}$ (see Methods for details). For $\frac{\nu^o}{\nu} = 0.01$, excess density builds up at the vortex core and can dominate over the usual density depletion in a purely inertial vortex (for which $\frac{\nu^o}{\nu} = 0$). Plotted as red dashed lines are the numerical solutions for the density profile near the vortex center in the presence of a moderately small antisymmetric stress. In these solutions, the density profile is modified, but the conclusion about the possible presence of a density peak at the vortex center does not change

$\nu^o$ is not associated with energy dissipation[27]. Significantly, the derivation presented in the Methods section for the odd viscosity of a chiral active gas relies only on conservation laws: it does not require dissipation.

As in ordinary hydrodynamics, two familiar dimensionless parameters can be used to classify different phenomena described by Eq. (5): (i) the Reynolds number $\mathrm{Re} \equiv v_0 r_0 / \nu$, where $r_0$ is the characteristic length scale associated with the initial flow profile and (ii) the Mach number $\mathrm{Ma} \equiv v_0 / c$, where the speed of sound $c$ enters via $c \equiv \sqrt{\partial p / \partial \rho}$. In the presence of odd viscosity, we need an additional dimensionless parameter: either the viscosity ratio $\nu^o / \nu$ or the odd Reynolds number $\mathrm{Re}^o \equiv v_0 r_0 / \nu^o$.

**Weak compressibility: vortices**. When the flow is incompressible, the odd viscosity can be absorbed by redefining the pressure: $p \to p - 2\eta^o \omega$[27]. Note that the reabsorption of odd viscosity can be done only in the equations of motion and not necessarily in the boundary conditions[31]. At low Mach and Reynolds numbers, we show that odd viscosity can replace inertia to stabilize a vortex with a density peak at its core. Vortices in active fluids are a ubiquitous phenomenon, also analyzed theoretically and experimentally in refs. [2,11,18,20,21,38]. However, much remains to be explored about the unique dynamics of vortices that results from activity. Studying the effects of $\eta^o$ on sustained vortex flow may be a viable route to measure odd viscosity in table-top experiments in classical driven or active fluids. In Fig. 2a, b, we plot the Lamb–Oseen vortex flow profile[15] (which does not depend on the value of odd viscosity), obtained from Eq. (5), and the profile of density variations at the center of the vortex. In the presence of odd viscosity, density deviates from the Lamb–Oseen profile (see Methods). The limit $\mathrm{Re}^o$ much less than 1 is the extreme case in which odd viscosity completely dominates over the inertial vortex density dip. In this limit, the density variations $\delta\rho \equiv \rho(x, t) - \rho_0$, measured relative to its constant value $\rho_0$ away from the vortex core, are controlled by the odd Reynolds number and are given by:

$$\frac{\delta\rho}{\rho_0} = \frac{2\mathrm{Ma}^2}{\pi(1 + t/t_0)\mathrm{Re}^o} e^{-\frac{r^2}{4\nu(t_0 + t)}}, \qquad (7)$$

where $t_0 \equiv r_0^2/(4\nu)$. We plot the rescaled form of Eq. (7) in Fig. 2c. The sign of the relative density at the vortex center depends on the relative sign of fluid vorticity and odd viscosity. In Fig. 2d, we show that for the case in which the rotational handedness of the vortex is aligned with the spinning direction of the rotors (i.e., $\mathrm{Re}^o > 0$), the center of the vortex experiences an increase in pressure and a resulting excess particle density even when the effects of inertia and a moderately small antisymmetric stress are non-negligible. (In the Methods, we show that the corrections due to inertia can be accounted for analytically, and derive the expression plotted in black solid lines in Fig. 2d). Contrast this scenario with the Lamb–Oseen solution ($\nu^o \to 0$), in which inertia causes a pressure dip and particle depletion at the vortex center (Fig. 2d). This effect, familiar from the physics of cyclones, is amplified when $\mathrm{Re}^o < 0$. In Fig. 2d, we also numerically compute vortex dynamics in the presence of antisymmetric stress (i.e., the full hydrodynamic Eqs. (2) and (3), for details see Methods), starting from an initial Lamb–Oseen profile. We find that although the addition of moderately small antisymmetric stress quantitatively corrects the flow and density profiles, the relative rotational handedness of the vortex still determines the characteristic density peak or trough at the vortex center.

**Strong compressibility: shocks**. At high Mach number, we consider strong effects of compressibility and obtain analytical and numerical solutions for modified Burgers shocks in compressible chiral active fluids. In ordinary fluids, where $\nu^o = 0$, Burgers shocks propagate along the direction of compression ($\hat{x}$ in Fig. 3) and are stabilized by the balance between dissipation and non-linearities in the convective derivative of Eq. (5). We find that in the presence of odd viscosity, ultrasonic shocks contain an additional flow transverse to the direction of shock propagation (Fig. 1c). Using the exact solution of the one-dimensional Burgers equation, we find an analytical expression for transverse flow $v_y(\hat{x})$, where $\hat{x} = x v_0 / 2\nu$ in the regime of low viscosity ratio (see Fig. 3a and Methods for

    

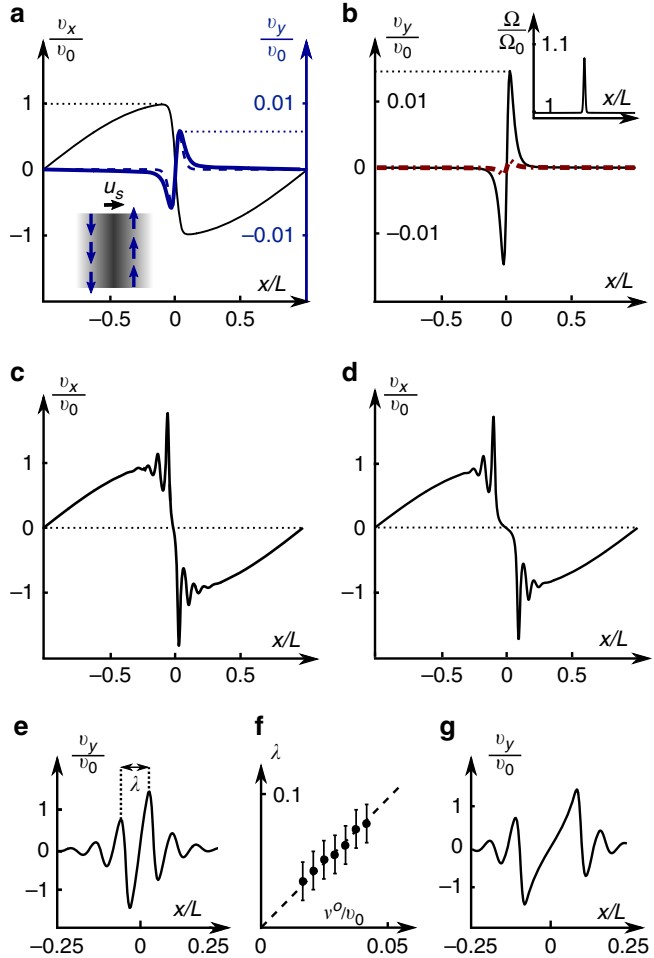

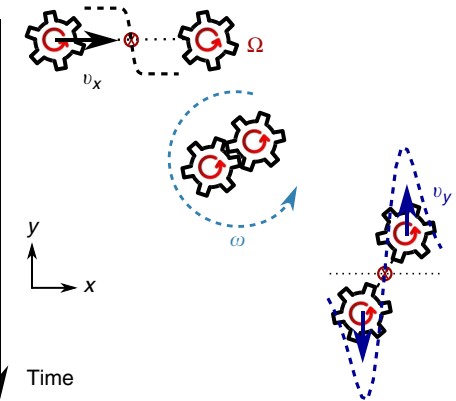

**Fig. 4** An intuitive picture for the shock profile. With the shock propagating at speed $v_x$, the left particle collides into the right one. During the collision, intrinsic rotation $\Omega$ is converted into the orbital rotation $\omega$ within the shock. After the collision, transverse flow has a profile with amplitude $v_y \sim a\Omega$, where $a$ is the particle size. This picture agrees well with the hydrodynamic description of the shape of transverse flow and the scaling of $v_y$

**Fig. 3** Shocks in chiral active fluids. **a** Inset: schematic of transverse flow $v_y$ (blue arrows) in a compression shock. Main panel: longitudinal [black, thin] and transverse [blue: analytic (dashed), numeric (solid, thick)] flow profiles for small viscosity ratio $\nu^o/\nu = -0.02$ in a stationary shock which results from symmetric external forcing $\mathbf{f}(x) = -\hat{x}\frac{f_0\pi}{4}\sin\pi x/L$. The characteristic velocity scale is $v_0 \equiv \sqrt{f_0 L/\rho_0}$; transverse flow $v_y$ scales as $v_0\frac{\nu^o}{\nu}$. **b** Even in the presence of antisymmetric stress, the effects of odd viscosity can dominate if the variation in $\Omega$ is small: $|\Omega - \Omega_0|/\Omega_0 \lesssim 10\%$ (inset: $\Omega/\Omega_0$). The sharper peak (black, solid) includes the combined effect of antisymmetric stress and odd viscosity. Neglecting odd viscosity, we find only a small transverse flow due to antisymmetric stress (red, dashed). **c**–**g** The longitudinal (**c**, **d**) and transverse (**e**, **g**) flow for $\nu^o/\nu = -10$. Shocks develop oscillations with wavelength $\lambda \sim |\nu^o|/v_0$: **f** numerical verification of this scaling law. In **c**, **e** and **f**, the antisymmetric stress is zero, whereas in **d** and **g** its value is as in **b**. See Supplementary Movie 1 for the numerical simulations that show the dynamics of relaxation to this steady state

transport equation for the vorticity $\omega$ is modified as (see Methods):

$$\partial_t\omega + \nabla\cdot(\omega\mathbf{v}) = \nu\nabla^2\omega + \frac{\nu^o}{2}\nabla^2(\nabla\cdot\mathbf{v}), \qquad (9)$$

where the additional source term, proportional to $\nu^o$, vanishes if the fluid is incompressible, i.e., away from the shock where $\nabla\cdot\mathbf{v} \to 0$. Note that this mechanism of generation of vorticity is independent of the inter-rotor friction captured by the antisymmetric stress that was explicitly neglected in writing Eq. (9). If a moderately small $\Gamma$ is introduced, the transverse flow acquires gradients in $\Omega$ (inset of Fig. 3b), but only small quantitative corrections to the flow profile (Fig. 3b).

The illustration of particle collisions in Fig. 4 gives us an intuitive way to understand the defining features of the shock profile, valid in the limit of very thin shocks. First, we distill these features from Eq. (8) by rewriting $v_y(x) = \nu^o/w f(x/w)$, where $f$ is a dimensionless odd function. Away from the shock center, $v_y^c \equiv v_y(w) \sim \nu^o/w$. In the chiral active gas, $\nu^o = \ell/(2\rho) \sim a^2\Omega$, where the equality comes from the derivation of Eq. (5) and the scaling relation from the expression $\ell \sim \rho a^2\Omega$ for intrinsic angular momentum density, where $a$ is the particle radius. For a shock so thin that $w \sim a$, this predicts $v_y^c \sim \Omega a$: an expression in terms of the single-particle rotation rate and radius only. We now reproduce it from a microscopic model of the chiral active gas based on Fig. 1b and shown in Fig. 4 in the context of a shock. The left particle flows with the shock, whereas the right particle is ahead of the shock front. When the particles collide they roll without slipping for a brief moment and their intrinsic angular momentum $\Omega a^2$ is converted into orbital angular momentum $v_y^c a$. When the particles lose contact, they maintain this orbital motion, with $v_y^c \sim \Omega a$. Note that this microscopic result agrees well with both the shape and the amplitude of the transverse flow in the shock derived using the hydrodynamic theory with an odd viscosity term.

More striking phenomena occur in the fast-spinning regime in which odd viscosity dominates over dissipation, i.e., $|\nu^o|$ is much greater than $\nu$. In this case, the shock profile changes qualitatively (contrast Fig. 3a, c). The odd viscosity introduces strong dispersive effects that, in addition to dissipation, give rise to non-linear waves reminiscent of KdV-Burgers shocks[39,40].

derivation):

$$\frac{\nu}{2\nu_o v_0}v_y(\hat{x}) = -\hat{x} + \tanh\hat{x}\,(\ln|2\cosh\hat{x}|). \qquad (8)$$

Note that the characteristic width $w$ of the transverse flow profile scales as $\nu/v_0$, i.e., as the dissipative viscosity, whereas the flow amplitude scales as the viscosity ratio $\nu^o/\nu$. The analytical profile of transverse flow (dashed blue line) agrees well with numerical solutions of chiral-active-fluid hydrodynamics (solid blue line). Note that the transverse flow (in particular, vorticity) is localized within the shock where $\nabla\cdot\mathbf{v}$ is largest. The familiar nonlinear

    

Consistent with this interpretation, the numerically obtained profile for the transverse response in Fig. 3c–g can be characterized in terms of oscillations of wavelength $\lambda \sim |\nu^o|/\nu_0$ that decay over distance $\Lambda \sim |\nu^o|\lambda/\nu$ (see Fig. 3f for numerical results, Fig. 3d, f illustrate the small effect of the antisymmetric stress.) These scaling laws can be derived by combining the transverse and longitudinal components of the steady-state Burgers equation into a single non-linear ordinary differential equation (see Methods):

$$\frac{1}{2}u_x\left(u_x^2 - 1\right) + \left[(\text{Re}^o)^{-2} + \text{Re}^{-2}\right]\partial_x^2 u_x = \text{Re}^{-1}\partial_x u_x^2. \quad (10)$$

Upon linearizing Eq. (10) in terms of the transverse flow plofile $\delta u \equiv (v_x - v_0)/v_0$, we find an equation for a damped harmonic oscillator:

$$\left[(\text{Re}^o)^{-2} + \text{Re}^{-2}\right]\partial_x^2(\delta u) - 2\text{Re}^{-1}\partial_x\delta u + \delta u = 0. \quad (11)$$

In Eq. (11), the trajectory of the harmonic oscillator describes the shape of the shock profile: the velocity oscillations within the spatial profile of the shock correspond to harmonic motion, whereas the decaying profile of the envelope around the shock center corresponds to the oscillator relaxing toward the energy minimum. Upon comparing the first and last terms in Eq. (11), we find the characteristic wavelength

$$\lambda \sim \frac{\sqrt{\left[(\nu^o)^2 + \nu^2\right]}}{\nu_0} \quad (12)$$

Upon comparing the first and second terms in Eq. (11), we obtain the damping ratio, which corresponds to the decay length of the envelope:

$$\Lambda \sim \frac{\left[(\nu^o)^2 + \nu^2\right]}{\nu\nu_0}. \quad (13)$$

These expressions correctly reproduce the scaling laws for regimes in which $\nu^o/\nu$ is either much greater than or much less than 1.

## Discussion

Two-dimensional incompressible inviscid fluids composed of many interacting vortices have been previously proposed as examples of emergent odd viscosity fluids[29]. However, unlike particles with active rotations, vortices in real fluids are not stable steady-state constituents, unless their circulation is quantized or viscosity is zero (as in superfluids). In this study, we have provided a derivation of the hydrodynamics of compressible active fluids with an emphasis on the effects of compressibility and nonlinearities. To summarize, chiral active fluids differ from electron fluids because they exhibit an odd viscosity that arises only out of equilibrium, is always accompanied by an antisymmetric stress, and is not well defined as particles jam and active rotations are hindered by interactions. Our results show that a chiral active fluid carries a crank mechanism within itself—it can convert between linear and rotational motion. We envision that this collective mechanical response could be exploited in self-assembled hydraulic devices and microscopic machines based on chiral active components.

## Methods

**Variational principle for hydrodynamics with odd viscosity**. To arrive at our results, we started with Eqs. (1)–(4), which in the main text were introduced phenomenologically. In this section, we present a derivation, based on a hydrodynamic variational principle, of a two-dimensional fluid characterized by the density of intrinsic angular momentum $\ell$. We assume that the internal energy density of the fluid $\epsilon(\rho, s, \ell)$ depends on the mass density $\rho$, entropy density $s$, and

the density of intrinsic angular momentum $\ell$. Standard thermodynamic formulae give the following relations between $\epsilon(\rho, s, \ell)$, pressure $p$, chemical potential $\mu$, temperature $T$, particle mass $m$, and intrinsic angular velocity $\Omega$:

$$p = \rho\epsilon_\rho + s\epsilon_s + \ell\epsilon_\ell - \epsilon, \quad (14)$$

$$\mu/m = \epsilon_\rho, \quad (15)$$

$$T = \epsilon_s, \quad (16)$$

$$\Omega = \epsilon_\ell. \quad (17)$$

Here, $\epsilon_\rho \equiv \partial\epsilon/\partial\rho$, $\epsilon_s \equiv \partial\epsilon/\partial s$, and $\epsilon_\ell \equiv \partial\epsilon/\partial\ell$.

Now, we consider the fluid in local equilibrium characterized by space- and time-dependent fields $\rho(x, t)$, $s(x, t)$, $\ell(x, t)$, and fluid velocity $v_i(x, t)$ ($i = 1, 2$). Based on the thermodynamic equations of state, we construct a hydrodynamic action for a fluid with intrinsic angular momentum $\ell$, given by:

$$S = -\int d^2x\,dt\left[\xi_0 + v_i\xi_i - \frac{\rho v_i v_i}{2} + \epsilon(\rho, s, \ell) - \omega\ell\right], \quad (18)$$

where the fluid vorticity $\omega$ is defined via $\omega \equiv \frac{1}{2}\epsilon_{ij}\partial_i v_j$. Note the unusual proportionality factor of 1/2 in the definition of vorticity, which we use throughout the work. The vorticity $\omega$ defined in this way is the local angular velocity of the rotating fluid. In Eq. (18), we used the notation:

$$\xi_\mu \equiv \rho\partial_\mu\theta + s\partial_\mu\eta + \ell\partial_\mu\phi + \Phi_\alpha\partial_\mu\Psi_\alpha \quad (19)$$

with $\mu = 0, 1, 2$. The action in Eq. (18) should be considered as a functional of the independent fields $\rho, \theta, s, \eta, \ell, \phi, \Phi_\alpha, \Psi_\alpha$, and $v_i$. The parameters $\rho, \theta, s, \eta, \ell$, and $\phi$ are called Clebsch parameters[36,37]. We have added auxiliary pairs of Clebsch parameters $\Phi_\alpha$ and $\Psi_\alpha$ with $\alpha = 1, 2, \ldots$ which are useful to describe generic hydrodynamic flows but will not play any role in the hydrodynamic equations.

The action in Eq. (18) is standard except for the last term. This term explicitly breaks parity as it depends on $\epsilon_{ik}$. Below, we show that the last term in Eq. (18) leads to a non-vanishing odd viscosity. The physical manifestation of this term is the main subject of this paper.

Varying over fields $\theta$, $\eta$, and $\phi$ we obtain the following conservation laws

$$\partial_t\rho + \partial_i(\rho v_i) = 0, \quad (20)$$

$$\partial_t s + \partial_i(s v_i) = 0, \quad (21)$$

$$\partial_t\ell + \partial_i(\ell v_i) = 0. \quad (22)$$

The velocity field $v_i$ in Eq. (18) is not a dynamical field. Varying Eq. (18) with respect to $v_i$ relates $v_i$ to Clebsch parameters from Eq. (19):

$$\rho v_i = \xi_i - \frac{1}{2}\partial_i^*\ell. \quad (23)$$

Varying Eq. (18) with respect to all other fields, after some manipulations, one arrives at the following equation of motion:

$$\partial_t(\rho v_i) + \partial_j\left[\rho v_i v_j + p\delta_{ij} - \sigma_{ij}^{\text{odd}}\right] = 0 \quad (24)$$

with

$$\sigma_{ij}^{\text{odd}} = \eta^o\left(\partial_i v_j^* + \partial_i^* v_j\right), \quad (25)$$

where

$$\eta^o = \frac{1}{2}\ell(\mathbf{x}, t), \quad (26)$$

which is the main result of this section. Note that $\eta^o$ in Eq. (25) is given by the intrinsic angular momentum field $\ell(\mathbf{x}, t)$ via Eq. (26). The quantity $\eta^o$ reduces to an anomalous transport coefficient (referred to as odd viscosity) when the angular momentum density $\ell$ is fixed (e.g., a constant). In general, interactions may generate higher-order corrections to the variational functional. However, the result in Eq. (26) holds exactly in the case of a weakly interacting chiral granular gas with fast local rotations. Here and in the following we use the notation $a_i^* \equiv \epsilon_{ik}a_k$. From Eq. (24) we identify the quantity

$$g_i \equiv \rho v_i \quad (27)$$

as a momentum density of the fluid and $\sigma_{ij}^{\text{odd}}$ as a part of a viscous stress tensor of the fluid.

We note that the three Eqs. (20)–(22) and the two components of Eq. (24) give us five equations sufficient to determine five independent fields $\rho, s, \ell$, and the two

components of $v_i$. We define these equations as a complete system of hydrodynamic equations. The number of hydrodynamic fields (five) is smaller than the number of fields in the variational principle Eq. (18). This reduction is known as symplectic or Hamiltonian reduction.

Another important remark is that the Eqs. (20)–(22) and (24) are necessarily dissipationless as they are derived from the time-translational-invariant action in Eq. (18). Indeed, it is easy to derive from these equations the energy conservation law:

$$\partial_t\left[\epsilon + \frac{\rho v_k v_k}{2}\right] + \partial_i\left[(\epsilon + p)v_i + \frac{\rho v_k v_k}{2}v_i - \sigma_{ij}^{\text{odd}}v_j\right] = 0. \tag{28}$$

**Dissipation and gradient corrections.** Let us consider the derivation of the energy conservation in more details. We proceed as follows

$$\begin{aligned}
&\partial_t\left[\epsilon + \frac{\rho v_k v_k}{2}\right] + \partial_i\left[(\epsilon + p)v_i + \frac{\rho v_k v_k}{2}v_i + \sigma_{ij}v_j\right] \\
&= -\sigma_{ij}\partial_i v_j + \left(\epsilon_\rho - \frac{u_i^2}{2}\right)\left[\partial_t\rho + \partial_i(\rho v_i)\right] \\
&\quad + \epsilon_s[\partial_t s + \partial_i(s v_i)] + \epsilon_\ell[\partial_t \ell + \partial_i(\ell v_i)] \\
&\quad + v_i\left[\partial_t(\rho v_i) + \partial_j(\rho v_i v_j + \delta_{ij}p - \sigma_{ij})\right].
\end{aligned} \tag{29}$$

So far we did not use any equation of motion. Using Eqs. (20)–(22) and (24) in the right hand side and the form of odd viscosity tensor Eq. (25), we obtain the energy conservation Eq. (28). Our goal is to generalize hydrodynamic equations in the presence of dissipation in such a way that the entropy production is explicitly positive and the energy is conserved up to the loss due to the work of external forces.

We assume the following modified equations of motion

$$\partial_t\rho + \partial_i(\rho v_i + J_i^\rho) = 0, \tag{30}$$

$$\partial_t\ell + \partial_i(\ell v_i + J_i^\ell) = \tau - \epsilon_{ij}\sigma_{ij} - \Gamma^\Omega\Omega, \tag{31}$$

$$\partial_t(\rho v_i) + \partial_j\left[\rho v_i v_j + p\delta_{ij} - \sigma_{ij}\right] = -\Gamma_{ij}^v v_j, \tag{32}$$

$$\partial_t s + \partial_i(s v_i + J_i^s) = \frac{Q}{T}. \tag{33}$$

Here, $\tau$ is the density of external torque acting on intrinsic rotational degrees of freedom of the fluid, $J_i^{\rho,s,\ell}$ are gradient corrections to currents, $\Gamma^\Omega$ is the intrinsic angular momentum damping rate, and $Q$ is a heat production rate due to various friction forces. We consider the term $-\Gamma_{ij}^v v_j$ to be a linear-momentum damping term resulting from friction of the particles with a substrate, and therefore to have the form $\Gamma_{ij}^v = \Gamma^v\delta_{ij}$. In principle, $\Gamma_{ij}^v$ could also include a Lorentz-like component proportional to $\epsilon_{ij}$, for example in a high-Reynolds number flow as a result of a Magnus force. We do not consider the effects of these $\Gamma_{ij}^v$ components in this work. The combination of Eqs. (31) and (32) produces the conservation of angular momentum density

$$\begin{aligned}
&\partial_t(\epsilon_{ki}x_k\rho v_i + \ell) + \partial_j\left(\epsilon_{ki}x_k\left[\rho v_i v_j + p\delta_{ij} - \sigma_{ij}\right] + \ell v_j + J_j^\ell\right) \\
&= \tau - \Gamma^v\epsilon_{ki}x_k v_i - \Gamma^\Omega\Omega,
\end{aligned} \tag{34}$$

where the right-hand side is the density of net torque acting on the system due to external forces.

We now fix all constitutive relations up to the first order in gradients[41], which guarantees that the heat production rate $Q$ is non-negative. We obtain

$$J_i^\rho = -D^\mu\partial_i\left(\mu - \frac{v_k^2}{2}\right) + L^\mu\partial_i^*\left(\mu - \frac{v_k^2}{2}\right), \tag{35}$$

$$J_i^s = -D^T\partial_i T + L^T\partial_i^* T, \tag{36}$$

$$J_i^\ell = -D^\Omega\partial_i\Omega + L^\Omega\partial_i^*\Omega, \tag{37}$$

and

$$\sigma_{ij} = \sigma_{ij}^s + \frac{1}{2}\sigma^a\epsilon_{ij}, \tag{38}$$

$$\sigma^a = \Gamma\left(\Omega - \frac{1}{2}\partial_k v_k^*\right), \tag{39}$$

$$\begin{aligned}
\sigma_{ij}^s &= 2\eta(\partial_i v_j + \partial_j v_i - \delta_{ij}\partial_k v_k) + \delta_{ij}\eta^b(\partial_k v_k) \\
&\quad + \eta^{\text{odd}}\left(\partial_i v_j^* + \partial_i^* v_j\right),
\end{aligned} \tag{40}$$

and for the total heat production rate

$$\begin{aligned}
Q &= D^\mu\left[\partial_i\left(\mu - \frac{v_k^2}{2}\right)\right]^2 + D^T(\partial_i T)^2 + D^\Omega(\partial_i\Omega)^2 \\
&\quad + \Gamma\left(\Omega - \frac{1}{2}\partial_k v_k^*\right)^2 \\
&\quad + \eta(\partial_i v_j + \partial_j v_i - \delta_{ij}\partial_k v_k)^2 + \eta^b(\partial_k v_k)^2.
\end{aligned} \tag{41}$$

Using these constitutive relations as well as Eqs. (29)–(33), we derive the modified energy conservation law

$$\begin{aligned}
&\partial_t\left[\epsilon + \frac{\rho v_k^2}{2}\right] + \partial_i\left[(\epsilon + p)v_i + \frac{\rho v_k^2}{2}v_i - v_j\sigma_{ji} + J_i^E\right] \\
&= \Omega\tau - \Gamma^v v_i^2 - \Gamma^\Omega\Omega^2.
\end{aligned} \tag{42}$$

The right-hand side is the energy influx through the work done by an external torque and the energy loss due to external frictional forces. The correction to energy current is given by

$$J_i^E = \left(\epsilon_\rho - \frac{v_k^2}{2}\right)J_i^\rho + TJ_i^s + \Omega J_i^\ell. \tag{43}$$

The system of hydrodynamic Eqs. (30)–(33) together with constitutive relations Eqs. (35)–(41) gives a very general hydrodynamic description of system of active rotors. This description is characterized by many phenomenological constants and is too general for our purposes.

In this paper we are looking for the effects related to the transport of angular momentum in the system of active rotors. Notably, we neglect thermal effects, all temperature dependences, and omit Eq. (33). We also choose a frame (a definition of velocity) such that $J_i^\rho = 0$ and neglect all "odd" coefficients $L^{\mu,T,\Omega} = 0$ except for $\Gamma$. This leaves us with a much simpler hydrodynamic theory, Eqs. (1)–(4).

**Hydrodynamics of chiral active fluids with odd viscosity.** Here, we start with Eqs. (1)–(4) of the main text:

$$\partial_t\rho + \partial_i(\rho v_i) = 0, \tag{44}$$

$$\partial_t\ell + \partial_i(\ell v_i) = \tau + D^\Omega\partial_i^2\Omega - \Gamma^\Omega\Omega - \epsilon_{ij}\sigma_{ij}, \tag{45}$$

$$\partial_t(\rho v_i) + \partial_j(\rho v_i v_j) = \partial_j\sigma_{ij} - \Gamma^v v_i, \tag{46}$$

and

$$\sigma_{ij} = -p\delta_{ij} + \eta\left(\partial_i v_j + \partial_j v_i - \delta_{ij}\partial_k v_k\right) + \eta^{\text{odd}}\left(\partial_i v_j^* + \partial_i^* v_j\right) + \frac{1}{2}\epsilon_{ij}\Gamma(\Omega - \omega). \tag{47}$$

We see from (45) that the anti-symmetric part of the stress $\sigma_{ij}^a = \epsilon_{ij}\Gamma(\Omega - \omega)/2$ has a meaning of an internal torque acting between intrinsic rotational degrees of freedom and the rotational motion of the fluid. In Eq. (46), we have subsumed the pressure term into the definition of the fluid stress tensor. For the case $\Gamma^v = 0$, Eqs. (44)–(46) are identical to Eqs. (1)–(3) in the main text.

The complete equations that we simulate numerically, derived in the appropriate regimes in the next two sections are the Eqs. (44)–(46) with

$$\sigma_{ij} = -p\delta_{ij} + \eta\left(\partial_i v_j + \partial_j v_i - \delta_{ij}\partial_k v_k\right) + \eta^{\text{odd}}\left(\partial_i v_j^* + \partial_i^* v_j\right) + \frac{1}{2}\epsilon_{ij}\Gamma(\Omega - \omega). \tag{48}$$

that capture the effects of both odd viscosity $\eta^{\text{odd}}$ and antisymmetric stress $\Gamma$.

Equations (44)–(46) together with (47) (we will also put bulk viscosity $\eta^b = 0$ and identify $\eta^{\text{odd}} = \ell/2$ in the following) is our starting point to study the effects of odd viscosity and anti-symmetric stress $\sigma^a$ on the dynamics of active rotors. The equations above are constructed phenomenologically. For a realistic system of active rotors one should either derive or at least estimate the values of various hydrodynamic parameters from the microscopic model.

The angular momentum density is given by $\ell \equiv I\Omega$ with $I \equiv \iota\rho$ as the moment of inertia density ($\iota \sim a^2$, where $a$ is the linear size of the fluid's constituents particles). The active torque density $\tau$ is proportional to the density of rotors. It injects energy into the fluid, thus breaking detailed balance. The friction-coefficient $\Gamma^\Omega$ saturates energy injection so as to allow the fluid to reach a non-equilibrium steady state, whereas $D^\Omega$ controls diffusion of local rotations.

The crucial ingredient in Eqs. (44)–(46) is the anti-symmetric part of the stress $\sigma_{ij}^a = \epsilon_{ij}\Gamma(\Omega - \omega)/2$ that couples the flow and local rotation degrees of freedom. Here $\omega = \frac{1}{2}\partial_k v_k^*$ is the local angular velocity of the rotation of the fluid. The coupling between $\Omega$ and $v$ through $\sigma^a$ terms of (50, 51) respects the corresponding Onsager relation and, therefore, does not inject additional energy into the system. These terms act only as frictional inter-rotor couplings that convert angular momentum between local rotation of the fluid particles and vorticity due to center-of-mass motion. This part is the same as in ref. [11], but in addition to the terms in

ref. [11] we also include the odd viscosity part of the stress tensor (47), proportional to the intrinsic angular momentum: $\eta^{\mathrm{odd}} = \ell/2$.

We may also derive Eqs. (44)–(46) by introducing the functional $F[\mathbf{v}, \Omega]$ as

$$F[\mathbf{v}, \Omega] = \int d\mathbf{x} \left\{ \frac{\Gamma^\nu}{2} v_i^2 + \frac{\Gamma^\Omega}{2} \Omega^2 + \frac{D^\Omega}{2} (\partial_i \Omega)^2 \right\}. \tag{49}$$

Using this non-negative functional, we can rewrite (45,46) as

$$\partial_t \ell + \partial_i (\ell v_i) = \tau - \sigma^a - \frac{\delta F}{\delta \Omega}, \tag{50}$$

$$\partial_t (\rho v_i) + \partial_j \left( \rho v_i v_j - \sigma_{ij}^s \right) = \frac{1}{2} \epsilon_{ij} \partial_j \sigma^a - \frac{\delta F}{\delta v_i}. \tag{51}$$

**Incompressible hydrodynamics of chiral active fluids**. In this section, we consider the case in which the fluid is nearly incompressible. We first solve the equation of motion for the velocity field by using the incompressibility condition $\nabla \cdot \mathbf{v} = 0$ and then substitute this result to find how the pressure $p$ deviates away from its steady-state value $p_0$. We then find the deviations in density by assuming small, linear compression, $\rho - \rho_0 = c^{-2}(p - p_0)$. This approximation is valid as long as $c^{-2}|p - p_0| \ll \rho_0$.

Numerically, we keep the antisymmetric stress term to compare its effects with the effects of odd viscosity. Then, we solve the following system of equations for both $\mathbf{v}$ and $\Omega$:

$$\rho_0 \partial_t v_i + \rho_0 \nabla \cdot (v_i \mathbf{v}) = \eta \nabla^2 v_i + \frac{I}{2} \partial_j \left[ \Omega \left( \partial_i v_j^* + \partial_i^* v_j \right) \right] - \partial_i p + \frac{\Gamma}{2} \epsilon_{ij} \partial_j [\Omega - \omega], \tag{52}$$

$$I \partial_t \Omega + I \nabla \cdot (\Omega \mathbf{v}) = D^\Omega \nabla^2 \Omega - \Gamma^\Omega \Omega + \tau - \Gamma[\Omega - \omega]. \tag{53}$$

We divide Eq. (52) by $\rho_0$ and Eq. (53) by $I$ and obtain

$$\partial_t v_i + \nabla \cdot (v_i \mathbf{v}) = \nu \nabla^2 v_i - \rho_0^{-1} \partial_i p + \frac{\iota}{2} \partial_j \left[ \Omega \left( \partial_i v_j^* + \partial_i^* v_j \right) \right] + \frac{\Gamma'}{2} \epsilon_{ij} \partial_j [\Omega - \omega], \tag{54}$$

$$\partial_t \Omega + \nabla \cdot (\Omega \mathbf{v}) = D^{\Omega'} \nabla^2 \Omega - \Gamma^{\Omega'} \Omega + \tau' - \Gamma' \iota^{-1} [\Omega - \omega] \tag{55}$$

where $\nu \equiv \eta/\rho_0$ is the kinematic viscosity, $\Gamma' \equiv \Gamma/\rho$, $D^{\Omega'} \equiv D^\Omega/I$, $\Gamma^{\Omega'} \equiv \Gamma^\Omega/I$, $\tau' \equiv \tau/I$, and $\iota \equiv I/\rho_0$.

Dimensionless parameters: In the following, we will be solving Eqs. (54) and (55) numerically. To understand better various regimes and to make further analytic progress it is useful to understand the dimensionless parameters governing the motion of the fluid.

Let us rescale all physical quantities to obtain a dimensionless problem. From initial conditions of the form $\mathbf{v} = v_0 \mathbf{u}(t = 0, \mathbf{r}_r \equiv \mathbf{r}/r_0)$, we obtain natural length and velocity scales. Then, we define dimensionless quantities (denoted by subscript r) via $\omega = \omega_r v_0/r_0$, $t = t_r r_0^2/\nu$, $p = p_r \nu v_0 \rho_0/r_0$, $\Omega = \Omega_r \Omega_0$, where $\Omega_0 \equiv \tau/\Gamma^\Omega = \tau'/\Gamma^{\Omega'}$ and find dimensionless equations,

$$\partial_t u_i + \frac{v_0 r_0}{\nu} \nabla \cdot (u_i \mathbf{u}) = \nabla^2 u_i - \partial_i p_r + \frac{\iota \Omega_0}{2\nu} \partial_j \left[ \Omega_r \left( \partial_i u_j^* + \partial_i^* u_j \right) \right] \\ + \frac{\Gamma'}{2\nu} \frac{\Omega_0 r_0}{v_0} \epsilon_{ij} \partial_j \Omega_r - \frac{\Gamma'}{2\nu} \epsilon_{ij} \partial_j \omega_r, \tag{56}$$

$$\partial_t \Omega_r + \frac{v_0 r_0}{\nu} \nabla \cdot (\Omega_r \mathbf{u}) = \frac{D^{\Omega'}}{\nu} \nabla^2 \Omega_r + \frac{\Gamma^{\Omega'} r_0^2}{\nu} (1 - \Omega_r) - \frac{\Gamma'}{\nu} \frac{r_0^2}{\iota} \Omega_r + \frac{\Gamma'}{\nu} \frac{r_0^2}{\iota} \frac{v_0}{\Omega_0 r_0} \omega_r, \tag{57}$$

where all derivatives are now dimensionless, $v_0 r_0/\nu = \text{Re}$ is the (small) Reynolds number, $\frac{\iota \Omega_0}{\nu} = 2\nu_o/\nu$, $\frac{\Gamma'}{2\nu} \frac{\Omega_0 r_0}{v_0}$ determines the coupling of the velocity field to gradients in the $\Omega$ field due to the antisymmetric stress term, $\frac{\Gamma'}{2\nu}$ determines the corrections to the dissipative viscosity due to the $\Omega$ field, $\frac{D^{\Omega'}}{\nu}$ determines the relative importance of diffusivity for $\Omega$, $\frac{\Gamma^{\Omega'} r_0^2}{\nu}$ determines the strength of the friction for $\Omega$, $\frac{\Gamma'}{\nu} \frac{r_0^2}{\iota}$ changes this friction via the antisymmetric stress, $\frac{\Gamma'}{\nu} \frac{r_0^2}{\iota} \frac{v_0}{\Omega_0 r_0}$ couples the vorticity to the $\Omega$ field.

Integrating out the spinning frequency in the chiral gas regime: Although we solve Eqs. (54) and (55) directly in our numerical computations, to make analytical progress, we consider the regime in which $\Omega \approx$ const. For this regime, the odd viscosity term dominates over the antisymmetric stress term.

We assume that the angular velocity of intrinsic rotations is $\Omega_0 \sim \tau/\Gamma^\Omega$ and that the parameter $\Gamma$ is sufficiently small so that there is a sufficiently large period of time over which one can consider particles rotating quickly and with almost no exchange between their intrinsic rotations and their center of mass motion. More precisely, we assume that $\omega \ll \Omega$ and comparing the odd viscosity term $\sim \eta^{\mathrm{odd}} \nabla^2 \mathbf{v}^*$ and the antisymmetric stress terms $\sim \Gamma \nabla^*(\Omega - \omega)$ we require $\eta^{\mathrm{odd}} \frac{v_0}{r_0} \gg \Gamma\Omega$ where $v_0$ and $r_0$ are typical velocity and length scales of the problem.

Taking $\eta^{\mathrm{odd}} \sim I\Omega$ we obtain the condition

$$\frac{\Gamma}{I} \ll \frac{v_0}{r_0}. \tag{58}$$

In addition, for Eq. (58) to be valid for all times, the antisymmetric stress term $\Gamma\omega$ in Eq. (55) must be smaller than the active torque $\tau$. Otherwise, the condition in Eq. (58) would be violated due to the coupling between intrinsic rotation and vorticity, and large spatial variations in $\Omega$ would occur. We require, therefore

$$\frac{\tau}{\Gamma} \gg \frac{v_0}{r_0}. \tag{59}$$

The necessary condition for the existence of a regime with both Eqs. (58) and (59) satisfied is the following relation between hydrodynamics parameters:

$$\tau I \gg \Gamma^2. \tag{60}$$

In this regime, we can assume that $\Omega \approx \Omega_0 = $ const. For this case and for (nearly incompressible) flows at low Mach number, Eq. (54) reduces to a modified Navier–Stokes equation with the addition of an odd viscosity term:[27]

$$\partial_t v_i + (\mathbf{v} \cdot \nabla) v_i = \nu \nabla^2 v_i + \nu_o \nabla^2 \epsilon_{ij} v_j - \frac{1}{\rho_0} \partial_i p, \tag{61}$$

i.e., Eq. (5) of the main text. Note that in the incompressible case, the effects of odd viscosity can be subsumed by redefining the pressure via $p^{\mathrm{eff}} \equiv p - 2\eta^o \omega$. We arrive at this conclusion by noting that $\nabla \cdot \mathbf{v} = 0$ implies $\nabla^2 \epsilon_{ij} v_j = 2\partial_i \omega$ and substituting this equation into Eq. (61). One can write

$$\partial_t \mathbf{v} + (\mathbf{v} \cdot \nabla) \mathbf{v} = \nu \nabla^2 \mathbf{v} - \nabla(p^{\mathrm{eff}}/\rho_0), \tag{62}$$

which is identical in form to the conventional Navier–Stokes equation (see, e.g., ref. [31]). One can find $p^{\mathrm{eff}}$ from the flow and find the pressure of the fluid from $p = p^{\mathrm{eff}} + 2\eta^o \omega$.

Outside of the limit defined by Eqs. (58) and (59), there are contributions due to antisymmetric stress whose effects we examine numerically. In the regime opposite to that set by Eqs. (58)–(60), in which the antisymmetric stress dominates, the odd viscosity term is a small nonlinear correction to the hydrodynamics and can be neglected for the same reasons as it was neglected in refs. [11,25].

Analytic solution for the Lamb–Oseen vortex with odd viscosity: Let us look for a radially symmetric vortex solution of Eq. (61). Taking the curl of Eq. (61), we obtain the equations for vorticity $\omega = \frac{1}{2}\nabla \times \mathbf{v}$:

$$\partial_t \omega + (\mathbf{v} \cdot \nabla)\omega = \nu \nabla^2 \omega, \tag{63}$$

This is a transport equation for vorticity with the diffusion-like term due to the shear viscosity. We consider an initial Gaussian vorticity profile, so that the azimuthal component of velocity is a function of the radius and the radial component is zero. Then, for initial conditions for the vorticity, we consider:

$$\omega(t = 0, r) = \frac{v_0}{r_0 \pi} e^{-r^2/r_0^2}, \tag{64}$$

In this case, the full solution is given by

$$\omega(t, r) = \frac{v_0 r_0}{4\pi\nu(t_0 + t)} e^{-r^2/4\nu(t_0 + t)}. \tag{65}$$

where $t_0 \equiv r_0^2/(4\nu)$. From this expression for $\omega$, we obtain the velocity profile satisfying the relations $\frac{1}{2}\nabla \times \mathbf{v} = \omega$ and $\nabla \cdot \mathbf{v} = 0$ and find

$$v_\theta(t, r) = \frac{v_0 r_0}{\pi r} \left[ 1 - e^{-r^2/4\nu(t_0 + t)} \right]. \tag{66}$$

This solution is identical to the conventional Lamb–Oseen solution as the odd viscosity does not enter Eq. (63). However, the resulting pressure is different.

We find the expression for the pressure from the radial equation of motion in polar coordinates:

$$\rho_0^{-1} \frac{\partial p^{\mathrm{eff}}}{\partial r} = \frac{v_\theta^2}{r}. \tag{67}$$

Using $p^{\mathrm{eff}} \equiv p - 2\eta^o \omega$, we obtain

$$p - p_\infty = 2\eta^o \omega + \rho_0 \int_{+\infty}^r dr' \frac{v_\theta^2(r')}{r'}, \tag{68}$$

where $p_\infty \equiv p(r = \infty)$. Introducing $r_s(t) = r_0 \sqrt{1 + t/t_0}$ and $p_s = \frac{\rho_0 v_0^2}{\pi}$, we have

$$p - p_\infty = \frac{p_s}{1 + t/t_0} \left[ \frac{1}{2\pi} \int_{+\infty}^{r^2/r_s^2} dq \frac{(1 - e^{-q})^2}{q^2} + \frac{2}{\text{Re}^o} e^{-r^2/r_s^2} \right],$$

where $\text{Re}^o \equiv r_0 v_0/\nu^o$ is an odd Reynolds number.

In the regime in which the Mach number $\mathrm{Ma} \equiv v_0/c \ll 1$, we use the relation $\rho - \rho_0 = c^{-2}(p - p_\infty)$ to calculate (small) changes in density as a result of the vortex flow:

$$\rho - \rho_0 = \frac{\rho_0 \mathrm{Ma}^2}{\pi^2 (1 + t/t_0)} \left[ \frac{1}{2} \int_{+\infty}^{r^2/r_s^2} dq \, \frac{(1 - e^{-q})^2}{q^2} + \frac{2\pi}{\mathrm{Re}^o} e^{-r^2/r_s^2} \right]. \tag{69}$$

Rescaling the density as

$$\delta\tilde{\rho} \equiv (\rho - \rho_0) \frac{\pi^2 (1 + t/t_0)}{\mathrm{Ma}^2} \tag{70}$$

and the radius as $\tilde{r} \equiv r/r_s$, we find that

$$\frac{\delta\tilde{\rho}}{\rho_0} = \left[ \frac{1}{2} \int_{+\infty}^{\tilde{r}^2} dq \, \frac{(1 - e^{-q})^2}{q^2} + \frac{2\pi}{\mathrm{Re}^o} e^{-\tilde{r}^2} \right]. \tag{71}$$

We plot this solution (a rescaled form valid for all times $t$) in Fig. 2c of the main text. In the case $\mathrm{Re}^o \ll 1$, we drop the first term of Eq. (71). Then, we plot $\delta\tilde{\rho}|\mathrm{Re}^o|/(2\pi\rho_0) = \mathrm{sign}(\nu^o) e^{-\tilde{r}^2}$ in the inset of Fig. 2c. This is also the regime in which Eq. (6) is valid.

Note that in addition to the condition $\mathrm{Ma} \ll 1$ that must be valid for these equations to hold, for small $\mathrm{Re}^o$ the self-consistency condition $\frac{|\rho - \rho_0|}{\rho_0} \ll 1$ also dictates that

$$\frac{\mathrm{Ma}^2}{|\mathrm{Re}^o|} \ll 1. \tag{72}$$

Notably, an odd viscosity fluid can be compressible even at low Mach number if the odd Reynolds number $|\mathrm{Re}^o|$ is sufficiently small, or equivalently, if the odd viscosity is sufficiently large.

In Fig. 2b of the main text, we plot the rescaled solution for the azimuthal velocity, i.e.,

$$\tilde{v}_\theta(\tilde{r})/v_0 = v_\theta(r) \pi r_s/r_0 = \left( 1 - e^{-\tilde{r}^2} \right)/\tilde{r}. \tag{73}$$

**Compression shocks in chiral active fluids.** In this section, we are interested in the conditions for which Eq. (46) reduces to Burgers' equation. Under these conditions, the fluid experiences a compression shock, but the density gradients are small. Schematically, we assume a lowest-order-nonlinearity and lowest-order-gradient expansion. From this assumption, we note that terms of the form $\nabla\rho\nabla\nu$ are higher order in nonlinearity than the viscosity terms $\nabla^2\nu$ and higher order in gradients than the nonlinear term $\nabla\nu^2$. Thus we may neglect these terms to derive a Burgers' equation with the addition of odd viscosity and antisymmetric stress. In that case, the full equations of motion are:

$$\partial_t \rho + \nabla \cdot (\rho \mathbf{v}) = 0, \tag{74}$$

$$\partial_t (\rho v_i) + \nabla \cdot (\rho v_i \mathbf{v}) = \eta \nabla^2 v_i - \partial_i p + \frac{\iota}{2} \partial_j \left[ \Omega \left( \partial_i v_j^* + \partial_i^* v_j \right) \right]$$
$$+ \frac{\Gamma}{2} \epsilon_{ij} \partial_j [\Omega - \omega] + f_i(\mathbf{r}), \tag{75}$$

$$\iota \partial_t (\rho\Omega) + \iota \nabla \cdot (\rho\Omega\mathbf{v}) = D^\Omega \nabla^2 \Omega - \Gamma^\Omega \Omega + \tau - \Gamma[\Omega - \omega], \tag{76}$$

where $f_i$ is the external forcing that gives rise to the steady-state shock. We rewrite the left-hand side of Eqs. (75) and (76) using the product rule and the continuity Eq. (74),

$$\partial_t \rho + \nabla \cdot (\rho \mathbf{v}) = 0, \tag{77}$$

$$\rho[\partial_t v_i + (\mathbf{v} \cdot \nabla) v_i] = \eta \nabla^2 v_i - \partial_i p + \frac{\iota}{2} \partial_j \left[ \Omega \left( \partial_i v_j^* + \partial_i^* v_j \right) \right]$$
$$+ \frac{\Gamma}{2} \epsilon_{ij} \partial_j [\Omega - \omega] + f_i(\mathbf{r}), \tag{78}$$

$$\iota\rho[\partial_t \Omega + (\mathbf{v} \cdot \nabla)\Omega] = D^\Omega \nabla^2 \Omega - \Gamma^\Omega \Omega + \tau - \Gamma[\Omega - \omega] \tag{79}$$

We divide both sides of Eq. (78) by $\rho$ and both sides of Eq. (79) by $\rho\iota$ and Taylor-expand the (dynamic) hydrodynamics coefficients in density, keeping only the lowest-order (constant) terms to find a set of equations similar to Eqs. (54) and

(55):

$$\partial_t \rho + \nabla \cdot (\rho \mathbf{v}) = 0, \tag{80}$$

$$\partial_t v_i + \nabla \cdot (v_i \mathbf{v}) = \nu \nabla^2 v_i - \rho^{-1} \partial_i p + \frac{\iota}{2} \partial_j \left[ \Omega \left( \partial_i v_j^* + \partial_i^* v_j \right) \right]$$
$$+ \frac{\Gamma'}{2} \epsilon_{ij} \partial_j [\Omega - \omega] + f_i'(\mathbf{r}), \tag{81}$$

$$\partial_t \Omega + \nabla \cdot (\Omega \mathbf{v}) = D^{\Omega'} \nabla^2 \Omega - \Gamma^{\Omega'} \Omega + \tau' - \Gamma' \iota^{-1} [\Omega - \omega], \tag{82}$$

where $f_i' = f_i/\rho$. Equations (80)–(82), along with the equation of state $p(\rho)$, describe the full hydrodynamics. In the limit of strong compression, we may drop the pressure term and Eqs. (81) and (82) then form a closed set,

$$\partial_t v_i + \nabla \cdot (v_i \mathbf{v}) = \nu \nabla^2 v_i + \frac{\iota}{2} \partial_j \left[ \Omega \left( \partial_i v_j^* + \partial_i^* v_j \right) \right] + \frac{\Gamma'}{2} \epsilon_{ij} \partial_j [\Omega - \omega] + f_i'(\mathbf{r}), \tag{83}$$

$$\partial_t \Omega + \nabla \cdot (\Omega \mathbf{v}) = D^{\Omega'} \nabla^2 \Omega - \Gamma^{\Omega'} \Omega + \tau' - \Gamma' \iota^{-1} [\Omega - \omega]. \tag{84}$$

After solving Eqs. (83) and (84), the density profile can be found using the continuity equation Eq. (80). We rescale these equations to find the dimensionless form. In the steady state, the characteristic length and velocity scales come from the forcing term $f_i'(\mathbf{r}) = (v_0^2/r_0) f_r'(\mathbf{r}/r_0)$. We also rescale $t = t_r r_0^2/\nu$, $\Omega = \Omega_r \Omega_0$ the same way as for Eqs. (56) and (57). We find a set of equations that describe the strongly nonlinear regime:

$$\partial_t u_i + \frac{v_0 r_0}{\nu} \nabla \cdot (u_i \mathbf{u}) = \nabla^2 u_i + \frac{v_0 r_0}{\nu} f_r'(\mathbf{r}_r) + \frac{\iota \Omega_0}{2\nu} \partial_j \left[ \Omega r \left( \partial_i u_j^* + \partial_i^* u_j \right) \right]$$
$$+ \frac{\Gamma'}{2\nu} \frac{\Omega_0 r_0}{v_0} \epsilon_{ij} \partial_j \Omega_r - \frac{\Gamma'}{2\nu} \epsilon_{ij} \partial_j \omega_r, \tag{85}$$

$$\partial_t \Omega_r + \frac{v_0 r_0}{\nu} \nabla \cdot (\Omega_r \mathbf{u}) = \frac{D^{\Omega'}}{\nu} \nabla^2 \Omega_r + \frac{\Gamma^{\Omega'} r_0^2}{\nu} (1 - \Omega_r) - \frac{\Gamma'}{\nu} \frac{r_0^2}{\iota} \Omega_r + \frac{\Gamma'}{\nu} \frac{r_0^2}{\iota} \frac{v_0}{\Omega_0 r_0} \omega_r. \tag{86}$$

Note that Eq. (86) is identical to Eq. (57), whereas Eq. (85) is Eq. (56), but with the forcing term instead of the pressure term. The other crucial feature of Eqs. (85) and (86) is that the incompressibility condition $\nabla \cdot \mathbf{u} = 0$ does not apply, unlike for Eqs. (56) and (57).

Given a forcing $\mathbf{f}_r'(\mathbf{r}_r) = (f_{rx}'(\mathbf{r}_r), 0)$, we find a steady state that depends only on the $x$-coordinate. This allows us to solve for the density profile using the relation $\partial_x \rho/\rho = -\partial_x v_x/v_x$ (from Eq. (80)), which along with the mean value $\rho_0$ of the density, allows us to calculate the profile $\rho(x)$ using either the numerical (Fig. 3a) or analytical (Fig. 3b–f) solutions for $v_x(x)$.

Shocks: integrating out the spinning frequency in the chiral gas regime: We now follow the same logic as was used in the previous section, but without assuming incompressibility. We use the same conditions on initial scales (58, 59). This regime is only possible when the condition on the hydrodynamic parameters (60) is satisfied. In this case, we can assume that $\Omega \approx \Omega_0 = \mathrm{const}$.

As in Sec. IV A2, outside of the limit defined by Eqs. (58) and (59), there are contributions due to antisymmetric stress whose effects we examine numerically. In the regime opposite to that set by Eqs. (58) and (59), in which the antisymmetric stress dominates, the odd viscosity term is a small nonlinear correction to the hydrodynamics and can be neglected for the same reasons as it was neglected in refs. [11,25].

In the case for which odd viscosity dominates, Eq. (83) reduces to a modified Burgers' equation with the additional odd viscosity term:

$$\partial_t v_i + (\mathbf{v} \cdot \nabla) v_i = \nu \nabla^2 v_i + \nu_o \nabla^2 \epsilon_{ij} v_j + f_i'(\mathbf{r}). \tag{87}$$

Equation (87) may be re-written as an equation for the evolution of vorticity, in which $\nu^o$ contributes an additional source term as a result of the compressible part $\nabla \cdot \mathbf{v}$ of the flow (provided that $\nabla \times \mathbf{f} = 0$):

$$\partial_t \omega + \nabla \cdot (\omega \mathbf{v}) = \nu \nabla^2 \omega + \frac{\nu_o}{2} \nabla^2 (\nabla \cdot \mathbf{v}). \tag{88}$$

As explained in the main text, the odd viscosity generates vorticity preferentially within the shock where gradients of $\nabla \cdot \mathbf{v}$ are largest. The dimensionless version of Eq. (87) reads (Re $= \frac{v_0 r_0}{\nu}$)

$$\partial_t u_i + \mathrm{Re}\,(\mathbf{u} \cdot \nabla) u_i = \nabla^2 u_i + \frac{\nu_o}{\nu} \nabla^2 \epsilon_{ij} u_j + \mathrm{Re}\, f_{ri}'(\mathbf{r}_r). \tag{89}$$

In the steady state, the one-dimensional profile for Eq. (89) is determined by

$$\mathrm{Re}\, u_x \partial_x u_x = \partial_x^2 u_x + \frac{\nu_o}{\nu} \partial_x^2 u_y + \mathrm{Re} f'_{rx}(x_r), \qquad (90)$$

$$\mathrm{Re}\, u_x \partial_x u_y = \partial_x^2 u_y - \frac{\nu_o}{\nu} \partial_x^2 u_x. \qquad (91)$$

Perturbative solution for small odd viscosity: For the case $\nu_o/\nu \ll 1$, we obtain a perturbative analytical solution for the steady-state shock profile by first neglecting the subdominant $\nu_o$ term in Eq. (90), solving this equation both within and outside the shock using matched assymptotics, and then substituting the solution inside the shock into Eq. (91). We choose the forcing to be $\mathbf{f}'_r(\mathbf{r}_r) = (-\pi \sin(x_r \pi)/4, 0)$ and solve the equation looking for the periodic solution with the period $x_r \in [-1, 1]$. In this paper, we solve for a stationary shock and not a propagating front, the later case might lead to a slightly different functional form but with the same essential structure. We solve

$$u_x \partial_x u_x = \nu_r \partial_x^2 u_x - \frac{\pi}{4} \sin(x_r \pi), \qquad (92)$$

where $\nu_r = \nu/(\nu_0 r_0) = \mathrm{Re}^{-1}$ is the corresponding inverse Reynolds number. For large Reynolds number ($\nu_r \ll 1$), the steady-state solution has a narrow shock in the vicinity of $x_r = 0$. Away from this region, the inertial term dominates and the steady-state velocity profile is obtained by integrating the expression $\partial_x u_x^2 = -\frac{\pi}{2} \sin(x_r \pi)$ to find the solution $u_x = -\mathrm{sign}(x_r) \cos(x_r \pi/2)$. Here we tuned the integration constant so that the shock is at $x_r = 0$ and the velocity is zero on average.

In the region $x_r/\nu_r \ll 1$, a different solution applies. In that region, Eq. (92) can be simplified by dropping the forcing term, and then integrated exactly using matched-asymptotic boundary conditions $u_x \to 1$ for $x_r/\nu_r \ll -1$, and $u_x \to -1$ for $x_r/\nu_r \gg 1$. The resulting solution is given by $u_x = -\tanh[x_r/(2\nu_r)]$. For the case $\nu_r \ll 1$ a simple interpolation,

$$u_x(x) = -\tanh[x_r/(2\nu_r)] \cos(x_r \pi/2) \qquad (93)$$

between the outer and the inner solution gives a reasonable approximation to the exact steady state over the entire range of values of $x_r$.

The solution for $u_y$ decays rapidly away from the shock, so to find the steady-state $u_y$ profile, it is sufficient to consider the inner solution for $u_x$. We substitute this solution into $[\nu_r = \nu^o/(\nu_0 r_0) = (\mathrm{Re}^o)^{-1}]$

$$u_x \partial_x u_y = \nu_r \partial_x^2 u_y - \nu_r^o \partial_x^2 u_x, \qquad (94)$$

and find a linear ODE for $u_y$:

$$\nu_r \partial_x^2 u_y + \tanh\left(\frac{x_r}{2\nu_r}\right) \partial_x u_y = \frac{\nu_r^o}{2\nu_r^2} \mathrm{sech}^2\left(\frac{x_r}{2\nu_r}\right) \tanh\left(\frac{x_r}{2\nu_r}\right). \qquad (95)$$

We multiply the above equation by $\cosh^2\left(\frac{x_r}{2\nu_r}\right)$ and integrate, which leads to:

$$\frac{du_y}{dx_r} = \frac{\nu_r^o}{\nu_r^2} \mathrm{sech}^2\left(\frac{x_r}{2\nu_r}\right) \ln\left| e^{-c_1} \cosh\left(\frac{x_r}{2\nu_r}\right) \right|, \qquad (96)$$

where $c_1$ is an arbitrary constant. We then integrate this expression again, to find

$$u_y(x_r) = -\frac{2\nu_r^o}{\nu_r}\left[\frac{x_r}{2\nu_r} - \tanh\left(\frac{x_r}{2\nu_r}\right)\left(\ln\left| e^{1-c_1} \cosh\left(\frac{x_r}{2\nu_r}\right) \right|\right)\right] + c_2. \qquad (97)$$

The boundary conditions $u_y \to 0$ as $x/(2\tilde{\nu}) \to \pm\infty$ require the choice of integration constants $c_1 = 1 - \ln 2$ and $c_2 = 0$. Substituting these values and simplifying, we find:

$$u_y(x_r) = -\frac{2\nu_r^o}{\nu_r}\left[\frac{x_r}{2\nu_r} - \tanh\left(\frac{x_r}{2\nu_r}\right)\left(\ln\left| 2\cosh\left(\frac{x_r}{2\nu_r}\right) \right|\right)\right]. \qquad (98)$$

We then re-express this solution in terms of the viscosity ratio $\nu_o/\nu$ and Reynolds number Re.

$$\frac{\nu}{2\nu_o} u_y(\hat{x}) = -\hat{x} + \tanh\hat{x}\left(\ln|2\cosh\hat{x}|\right), \qquad (99)$$

where $\hat{x} = x_r \mathrm{Re}/2$ ($= x\nu_0/2\nu$). We plot this solution in Fig. 3a alongside the numerical solution to the full Eq. (87) once the system reaches the steady state.

Scaling for large odd viscosity: For the case $|\nu_o|/\nu \gg 1$, we use scaling to obtain the characteristic features of the profile. The equations that determine the steady-state profile are:

$$u_x \partial_x u_x = \nu_r \partial_x^2 u_x + \nu_r^o \partial_x^2 u_y - \frac{\pi}{4} \sin(x_r \pi), \qquad (100)$$

$$u_x \partial_x u_y = \nu_r \partial_x^2 u_y - \nu_r^o \partial_x^2 u_x. \qquad (101)$$

Let us drop the forcing and $\nu$ terms. We obtain

$$u_x \partial_x u_x = \nu_r^o \partial_x^2 u_y, \qquad (102)$$

$$u_x \partial_x u_y = -\nu_r^o \partial_x^2 u_x. \qquad (103)$$

Integrating the first equation gives $2\nu_r^o \partial_x u_y = u_x^2 - C_1$, where $C_1 = 1$ for the case we consider. Substituting into the second equation we obtain

$$u_x(u_x^2 - 1) = -2(\nu_r^o)^2 \partial_x^2 u_x.$$

This equation describes the motion of a nonlinear pendulum. We now linearize in $\delta u = u_x - 1$ to find the harmonic oscillator equation

$$\delta u = -(\nu_r^o)^2 \partial_x^2(\delta u). \qquad (104)$$

The solutions to this equations oscillate in space with a period given by $\lambda \sim \nu_r^o$, which is the scaling law that we observe numerically in Fig. 3c.

If we take Eqs. (100) and (101) and drop the forcing term, but not the $\nu$ terms, performing the same operations of integration we obtain the equation

$$\nu_r^o \partial_x u_y = \frac{1}{2}(u_x^2 - 1) - \nu_r \partial_x u_x. \qquad (105)$$

Substituting for gradients of $u_y$ in Eq. (101), we then obtain

$$\frac{1}{2} u_x(u_x^2 - 1) + \left[(\nu_r^o)^2 + \nu_r^2\right] \partial_x^2 u_x = \nu_r \partial_x u_x^2. \qquad (106)$$

Note that this is a nonlinear damped oscillator, with the damping term on the right-hand side. Linearizing in $\delta u = u_x - 1$, we find

$$\left[(\nu_r^o)^2 + \nu_r^2\right] \partial_x^2(\delta u) - 2\nu_r \partial_x \delta u + \delta u = 0, \qquad (107)$$

which describe the motion of a damped harmonic oscillator. From this, we find the characteristic wavelength to be (from comparing first and last terms)

$$\lambda^2 \sim \left[(\nu^o)^2 + \nu^2\right]/\nu_0^2 \qquad (108)$$

and the decay length of the envelope from the damping ratio (from comparing first and second terms):

$$\Lambda \sim \left[(\nu^o)^2 + \nu^2\right]/\nu\nu_0. \qquad (109)$$

In the limit $|\nu_o|/\nu \gg 1$, the oscillation wavelength is given by $\lambda \sim |\nu^o|/\nu_0$ (same as above) and the envelope size is $\Lambda \sim (\nu^o)^2/\nu\nu_0 \sim |\nu^o|\lambda/\nu$. We find that these scaling laws are consistent with the full numerical solution of Eq. (87), see inset of Fig. 3c. As expected, the envelope size diverges as viscosity goes to zero. In the opposite limit, $|\nu_o|/\nu \ll 1$, also considered in the previous section, we recover the same scaling laws: $\lambda \sim \Lambda \sim \nu/\nu_0$. In that limit, the oscillator is critically damped and no oscillations are observed.

**Numerical simulations.** In this section, we discuss the numerical method used for the direct numerical simulations of hydrodynamic equations. The equations are solved using the pseudo-spectral method. This method involves calculating spatial derivatives in Fourier space and non-linear terms in real space. The spatial derivatives are non-local in real space and local in Fourier space, while the non-linearities are local in real space but non-local in Fourier space. Thus, in this method we restrict ourselves in evaluating local terms in both real and Fourier space. The time integration is done to the (spatially) Fourier-transformed fields. To calculate these Fourier transforms we use the open-source library fftw-2.1.5.

In order to understand the implementation of the algorithm let us look at the various kind of terms that need to be implemented. The Laplacian operator in Fourier space is given by $\nabla^2 \to -k^2$. Let us now consider the (advection like) non-linear terms of the form $\nabla \cdot (\rho \mathbf{v})$. The product $\rho \mathbf{v}$ is a convolution in Fourier space and highly non-local, but in real space this is a local product. Therefore, the product is calculated in real space and the divergence of the term evaluates to $ik_x \widehat{\rho v_x} + ik_y \widehat{\rho v_y}$.

Now let us consider an equation of the form $\partial_t \rho + \nabla \cdot (\rho \mathbf{v}) = D^\rho \nabla^2 \rho$. In order to solve this pseudo-spectrally, we take the spatial Fourier transform of the equation and we are left with $\partial_t \hat{\rho} + ik_x \widehat{\rho v_x} + ik_y \widehat{\rho v_y} = -D^\rho k^2 \hat{\rho}$. It is possible to further simplify the equation by rearranging the terms and calculating the integration factor. We are finally left with: $\partial_t e^{D^\rho k^2 t} \hat{\rho} = -e^{D^\rho k^2 t}(ik_x \widehat{\rho v_x} + ik_y \widehat{\rho v_y})$. The solution of the above equation is:

$$\hat{\rho}(t) = -e^{-D^\rho k^2 t} \int_{t_0}^t dt' e^{D^\rho k^2 t'} ik \cdot \widehat{\rho \mathbf{v}}. \qquad (110)$$

The time evolution is done using a second-order Runge–Kutta (RK2) method. The pseudo-spectral method has the additional advantage of having exponentially small spatial truncation error as opposed to algebraically small error, which would result

were we to use a finite-difference scheme. The time evolution error in RK2 scheme is of the order of $\sim \mathcal{O}(\delta t)^2$.

One of the key concerns on calculating the convolutions numerically is the generation of aliasing errors. In our simulations, we cut off wave numbers greater than $2K/3$ where $K \equiv N/2$ corresponds to the smallest lengthscales in our system ($N$ is the number of grid points).

Incompressible regime: We solve the incompressible Navier–Stokes equations coupled to the local rotation field. We define a new tensor $\phi_{ij} = \partial_i v_j^* + \partial_i^* v_j$. The incompressible Navier–Stokes can be further simplified using the constraint $\nabla \cdot \mathbf{v} = 0$. Taking the curl of Eq. (54). The resulting equations are:

$$\partial_t \omega + \partial_l (v_l \omega) = \nu \nabla^2 \omega + \frac{\iota}{2} \varepsilon_{ik} \partial_k \partial_j \left( \Omega \phi_{ij} \right) + \frac{\Gamma'}{2} \nabla^2 [\Omega - \omega]. \quad (111)$$

$$\partial_t \Omega + \partial_l (v_l \Omega) = D^{\Omega} \nabla^2 \Omega - \Gamma^{\Omega'} \Omega + \tau' - \Gamma' \iota^{-1} (\Omega - \omega). \quad (112)$$

The above equations can be written in the form discussed above in the Fourier space to be integrated over time numerically. The form of the solutions are:

$$\hat{\omega}(t) = -e^{-\nu k^2 t} \int_{t_0}^{t} dt' e^{\nu k^2 t'} \left[ i k_l \widehat{v_l \omega} - \frac{\Gamma'}{2} k^2 (\hat{\Omega} - \hat{\omega}) - \frac{\iota}{2} \varepsilon_{ij} k_k k_j \widehat{\Omega \phi_{ij}} \right], \quad (113)$$

$$\hat{\Omega}(t) = -e^{-\left(D^{\Omega'} k^2 + \Gamma^{\Omega'}\right) t} \int_{t_0}^{t} dt' e^{\left(D^{\Omega'} k^2 + \Gamma^{\Omega'}\right) t'} \left[ i k_l \widehat{v_l \Omega} + \frac{\Gamma'}{\iota} \left( \hat{\Omega} - \hat{\omega} \right) \right]. \quad (114)$$

Now, pressure can be calculated by taking the divergence of Eq. (54)

$$\rho_0^{-1} \nabla^2 p^{\text{eff}} = -\partial_i [\nabla \cdot (\mathbf{v} v_i)] + \frac{\iota}{2} \partial_i \partial_j \left( \Omega \phi_{ij} \right). \quad (115)$$

The above equations provide us the information presented in the Fig. 2b, d in the main paper. The dimensionless parameters of the simulations are $\left( \frac{v_0 r_0}{\nu}, \frac{\nu_o}{\nu}, \frac{\Gamma'}{2\nu}, \frac{\Omega_0 r_0}{v_0}, \frac{D^{\Omega}}{\nu}, \frac{\Gamma^{\Omega'} r_0^2}{\nu}, \frac{r_0^2}{\iota} \right) = (0.05, \pm 0.01, 0.25, 0.004, 1, 0.1, 1)$ on a grid of size $L/r_0 = 20\pi$ and lattice spacing $a/r_0 = 0.2$ over a time $\tilde{t} = 1.0$. As initial conditions we use: $\tilde{\omega} = \frac{e^{-r^2}}{\pi}$ and $\tilde{\Omega} = 0$.

Highly compressible regime: The two-dimensional Burgers' equation (Eqs. (75) and (76)) that is used to model compressible flow is similarly solved by the pseudo-spectral algorithm. The data for Fig. 3 are obtained by solving this equation. The integral form of the solutions in Fourier space is:

$$\hat{v}_i(t) = -e^{-\nu k^2 t} \int_{t_0}^{t} dt' e^{\nu k^2 t'} \left[ i k \cdot \widehat{\mathbf{v} v_i} + \nu_o k^2 \epsilon_{ij} \hat{v}_j + i \frac{\iota}{2} k_j \widehat{\Omega \phi_{ij}} + i \frac{\Gamma'}{2} \epsilon_{ij} k_j (\hat{\Omega} - \hat{\omega}) \right]. \quad (116)$$

The solutions for $\Omega$ is the same as that for the incompressible flow.

We simulate Eqs. (83) and (84) with an applied forcing $f_r'(\mathbf{r}) = \sin(x)/2$ for $x$ is in $[0,2\pi]$ (we rescale all lengths according to $x \to (x - \pi)/\pi$ in our results). We use parameters $\left( v_0 r_0/\nu, \nu_o/\nu, \frac{\Gamma'}{2\nu}, \frac{\Omega_0 r_0}{v_0}, \frac{D^{\Omega}}{\nu}, \frac{\Gamma^{\Omega'} r_0^2}{\nu}, \frac{r_0^2}{\iota} \right) = (44, -0.02, 2.0, 2.2, 2.0, 1800, 9)$ with lattice spacing $a = 0.006$. For the simulations with high odd viscosity ratio (Fig. 3c, d of the main paper), we use $\nu_o/\nu = -10$.

**Data availability**. The data that support the plots within this paper and other findings of this study are available from the corresponding author upon request.

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

## Acknowledgements

We thank D. Bartolo, S. Ganeshan, W. Irvine, A. Levine, G. Monteiro, P. Wiegmann, and F. Ye for stimulating discussions. A.S., D.B., and V.V. were funded by FOM, NWO (Vidi

grant), and the Delta Institute for Theoretical Physics. This work was partially supported by the University of Chicago Materials Research Science and Engineering Center, which is funded by the National Science Foundation under award number DMR-1420709. A.G. A. acknowledges the financial support of the NSF under grant no. DMR-1606591 and the hospitality of the Kadanoff Center for Theoretical Physics.

## Author contributions

D.B., A.S., A.G.A., and V.V. contributed to the design of the project, analytical calculations, and writing of the manuscript. D.B. performed the numerical simulations.

## Additional information

**Competing interests:** The authors declare no competing financial interests.

