## [Peer review file · Nature Communications]

Reviewers' comments:

Reviewer #1 (Remarks to the Author):

The manuscript reports a detailed theoretical study of the viscosity of chiral active fluids. These fluids behave very differently from ordinary fluids, as their active rotors continuously inject energy and angular momentum at the microscale. Therefore, the authors predict that these active fluids have a dissipationless linear-response coefficient called "odd viscosity", which can lead, inter alia, to a density increase in the core of a fluid vortex. Also, active fluids are predicted to show transverse fluid flows in shock propagation.

These results and predictions are very interesting, and demonstrate that active fluids have many unexpected properties. These are interesting from the point of view of non-equilibrium statistical physics, but also from the point of view of potential applications.

However, the authors should address the following comments and questions before a decision about the suitability of the manuscript for publication in Nature Communications can be made:

(1) The authors often refer to the SI for detailed derivations. This is fine, since a Nature Communications manuscript is not the place for detailed derivations. However, I think some heuristic arguments and explanations in the main text would be very helpful for the reader to understand the essential points of the theory and the derivations.

For example, it would be very nice if Eqs. (2), (4) and (5) could be motivated and explained in this way.

(2) I find some of the figures difficult to understand! What is the inset of Fig. 2b supposed to show? What do I see in the inset of Fig. 2c? Fig. 2d is essentially identical to Fig. 2c, and the additional red dashed lines in Fig. 2d could easily be transferred to Fig. 2c.

(3) What is the length scale r_0 ?

(4) The reference to some figures seems to be incorrect. For example, it seems to be that in the first line after Fig. 3, it should be "In Fig. 2c, we numerically ..." instead of "Fig. 3c".

(5) The authors emphasize that the last term in Eq. (4) has been neglected in previous hydrodynamic theories (Refs. [11,20-24]). It should be discussed in more detail which consequences arise from this additional term.

Reviewer #2 (Remarks to the Author):

The paper meets the criteria listed in your peer review policy. (Except that I would not call most papers in any journal, not even Nature, "Extremely important". Extremes, almost by definition, are rare events.)

The paper links an observation discovered in the context of the quantum Hall effect with the behavior of classical non-equilibrium physics of active matter. The authors derive the governing equations and predict several remarkable features of such systems. They also suggest practical applications.

I believe there is a typo in "In Fig 3c" ten lines below eq.6. It should be Fig. 2c.

I was also confused how the plot in fig 2c for zero odd viscosity is consistent with eq. 6.

Finally, I may add that I particularly enjoyed the supplementary material. This reflects my taste as a theorist: I enjoy and appreciate conceptual structures even more than potential applications.

Reviewer #3 (Remarks to the Author):

Odd viscosity was first discussed by Avron in the late 1990s as an abstract concept. When time reversal is broken, then the Onsager relation is not valid anymore and the viscosity tensor contains an antisymmetric contribution. The current manuscript now connects odd viscosity with a special case of active matter that is active chiral fluids, which has energy introduced into rotational motion via the local application of torque. That such systems follow some of the behaviors of odd viscosity (e.g. development of vortices; spontaneous appearance of rotational flow at density variations, e.g. interfaces) has been shown in the past but without the usage of the term 'odd viscosity' and without an in-depth theoretical investigation. The main contribution of the manuscript is the development of a hydrodynamic theory. This theory is derived and solved numerically in a few select cases.

Overall, I find the present work an interesting and important contribution to the field of active matter. It will be of relevance to researchers in soft and granular matter by calling attention to an unusual and lesser known fluid dynamics phenomenon. There are some inaccuracies and issues as discussed in detail below. The main text is written well, while the Supplementary Information is very technical and will be hard for all but a few expert readers to follow. I would have hoped for more insights into what odd viscosity can be used for.

1a) As far as I understand, there are two ways to introduce rotation into a fluid: 'internal' and 'external'. 'Internal' means particles rotate themselves by pushing on their neighbors or on the surrounding fluid. Examples for this situation are bacteria rotating by beating flagella or molecular motors. 'External' means there is a torque supplied from the outside of the system, e.g. via a magnetic or an electric field. Internal rotation conserves overall angular momentum while external rotation does not.

1b) I wonder, and this is not made clear, which part of the hydrodynamic theory applies to internal rotation and which part only to external rotation? The abstract and figure 1 suggests that both are relevant for the theory developed here, but I am not sure. A vortex like the one shown in Fig. 2a is certainly not possible with internal rotation. The authors also mention that conservation of angular momentum needs to be violated in order for odd viscosity to appear (first paragraph), which suggests that the theory might only apply to external rotation. Somewhere there seems to be a contradiction. In short: What is the definition of a chiral active fluid as used e.g. in the title? Does it comprise both types of rotation?

2) As the authors discuss on page 2, their theory focuses on rotors with high spinning frequency and without a surrounding fluid (dry fluid). Translational friction is also ignored ($\Gamma \cdot v = 0$, see end of first paragraph on page 2, top right). In this situation rotors quickly reach an equilibrium between applied torque and friction, and interaction of rotors are less important. It is my impression this excludes all systems but systems of driven grains in figure 1. Is this correct? Which experimental systems would the theory developed be applicable to?

3) Equation (5) of the manuscript appears identical to equation (41) of Avron (Ref. [13]). I thus wonder how much overlap of the current theory exists with the work in this reference? It is my impression the authors tackled a more general problem but I am a bit lost in the technical parts of the work presented in Supplementary Information, which covers more than 100 equations.

4) While the Supplementary Information is certainly helpful, I wonder if it is all needed? It appears to go significantly beyond the main text.

5) The build-up of vortices has already been seen in experiment and simulation (e.g. Wensink et al. <https://journals.aps.org/pre/abstract/10.1103/PhysRevE.89.010302>; Refs [2,17,19]).

6) Subfigures are not always discussed in the order of their appearance in the text. Some subfigures (2c, 2d) are not explicitly mentioned.

7) At the beginning of the Supplementary Information covariant notation is sometimes used. However, this is not done in the main text or the later parts of the Supplementary Information. I think the convention should be kept consistent.

8) Would it be helpful to include the numerical solutions as a video instead of only the sections shown in the graphs of Figure 3?

Reviewers' comments:

Reviewer #1 (Remarks to the Author):

The manuscript reports a detailed theoretical study of the viscosity of chiral active fluids. These fluids behave very differently from ordinary fluids, as their active rotors continuously inject energy and angular momentum at the microscale. Therefore, the authors predict that these active fluids have a dissipationless linear-response coefficient called "odd viscosity", which can lead, inter alia, to a density increase in the core of a fluid vortex. Also, active fluids are predicted to show transverse fluid flows in shock propagation.

These results and predictions are very interesting, and demonstrate that active fluids have many unexpected properties. These are interesting from the point of view of non-equilibrium statistical physics, but also from the point of view of potential applications.

However, the authors should address the following comments and questions before a decision about the suitability of the manuscript for publication in Nature Communications can be made:

We would like to thank the reviewer for their encouraging remarks. We have significantly modified the main text of the manuscript to address the comments and questions that the reviewer has raised.

(1) The authors often refer to the SI for detailed derivations. This is fine, since a Nature Communications manuscript is not the place for detailed derivations. However, I think some heuristic arguments and explanations in the main text would be very helpful for the reader to understand the essential points of the theory and the derivations. For example, it would be very nice if Eqs. (2), (4) and (5) could be motivated and explained in this way.

We wholeheartedly agree with this point and have added several passages to the main text in order to better motivate and explain the derivations presented in the SI. We have taken special care to motivate Eqs. (2), (4), and (5) in this way and now moved parts of the derivation of these equations from the SI to the Methods section of the main text. We have also included a passage and subfigure on an intuitive microscopic picture for odd viscosity in the context of a compression shock.

(2) I find some of the figures difficult to understand! What is the inset of Fig. 2b supposed to show? What do I see in the inset of Fig. 2c? Fig. 2d is essentially identical to Fig. 2c, and the additional red dashed lines in Fig. 2d could easily be transferred to Fig. 2c.

We have modified Fig. 2 and expanded the caption for both Fig. 2 and Fig. 3 to address these issues. To summarize: the inset in Fig. 2b is a schematic figure illustrating the coupling between vorticity and pressure due to odd viscosity.

(3) What is the length scale r_0 ?

The characteristic length scale of the vortex, r_0 , comes from the choice of initial conditions. We have added a brief clarification of this.

(4) The reference to some figures seems to be incorrect. For example, it seems to be that in the first line after Fig. 3, it should be "In Fig. 2c, we numerically ..." instead of "Fig. 3c".

We thank the reviewer for pointing out this mistake, which we have now corrected.

(5) The authors emphasize that the last term in Eq. (4) has been neglected in previous hydrodynamic theories (Refs. [11,20-24]). It should be discussed in more detail which consequences arise from this additional term.

This is an important point: the additional term in Eq. (4) is the origin of odd viscosity, which results from this term in the limit of the chiral active gas. All of the phenomenology we consider originates from the generic presence of this term in the chiral active fluids. We hope that our modifications of the manuscript text have clarified the origins and consequences of this term.

Reviewer #2 (Remarks to the Author):

The paper meets the criteria listed in your peer review policy. (Except that I would not call most papers in any journal, not even Nature, "Extremely important". Extremes, almost by definition, are rare events.)

The paper links an observation discovered in the context of the quantum Hall effect with the behavior of classical non-equilibrium physics of active matter. The authors derive the governing equations and predict several remarkable features of such systems. They also suggest practical applications.

We thank the reviewer for their positive feedback. We have addressed the points the reviewer has raised below and in the manuscript text.

I believe there is a typo in "In Fig 3c" ten lines below eq.6. It should be Fig. 2c.

We thank the reviewer for pointing out this mistake, which we have now corrected.

I was also confused how the plot in fig 2c for zero odd viscosity is consistent with eq. 6.

This is a subtle point: Fig. 2c and Eq. 6 are plotted in different regimes. In Eq. 6, inertia has been completely neglected, and a nonzero odd viscosity is necessary to stabilize a vortex. On the other hand, Fig. 2c is at a small, but nonzero Reynolds number, in which vortices contain the effects of both odd viscosity and inertia. In that case, if the odd viscosity is zero, we recover the well-known Lamb-Oseen vortex solution of the Navier-Stokes equations, which is plotted in Fig. 2c. We hope that the edits to the manuscript have made this subtlety more clear.

Finally, I may add that I particularly enjoyed the supplementary material. This reflects my taste as a theorist: I enjoy and appreciate conceptual structures even more than potential applications.

We appreciate the kind words!

Reviewer #3 (Remarks to the Author):

Odd viscosity was first discussed by Avron in the late 1990s as an abstract concept. When time reversal is broken, then the Onsager relation is not valid anymore and the viscosity tensor contains an antisymmetric contribution. The current manuscript now connects odd viscosity with a special case of active matter that is active chiral fluids, which has energy introduced into rotational motion via the local application of torque. That such systems follow some of the behaviors of odd viscosity (e.g. development of vortices; spontaneous appearance of rotational flow at density variations, e.g. interfaces) has been shown in the past but without the usage of the term 'odd viscosity' and without an in-depth theoretical investigation. The main contribution of the manuscript is the development of a hydrodynamic theory. This theory is derived and solved numerically in a few select cases.

Overall, I find the present work an interesting and important contribution to the field of active matter. It will be of relevance to researchers in soft and granular matter by calling attention to an unusual and lesser known fluid dynamics phenomenon. There are some inaccuracies and issues as discussed in detail below. The main text is written well, while the Supplementary Information is very technical and will be hard for all but a few expert readers to follow. I would have hoped for more insights into what odd viscosity can be used for.

We thank the reviewer for encouraging remarks. We have addressed the reviewer's points by substantially modifying the main text of the manuscript and adding a Supplementary Movie of the numerical solutions. We have detailed these modifications in the reply below.

1a) As far as I understand, there are two ways to introduce rotation into a fluid: 'internal' and 'external'. 'Internal' means particles rotate themselves by pushing on their neighbors or on the surrounding fluid. Examples for this situation are bacteria rotating by beating flagella or molecular motors. 'External' means there is a torque supplied from the outside of the system, e.g. via a magnetic or an electric field. Internal rotation conserves overall angular momentum while external rotation does not.

1b) I wonder, and this is not made clear, which part of the hydrodynamic theory applies to internal rotation and which part only to external rotation? The abstract and figure 1 suggests that both are relevant for the theory developed here, but I am not sure. A vortex like the one shown in Fig. 2a is

certainly not possible with internal rotation. The authors also mention that conservation of angular momentum needs to be violated in order for odd viscosity to appear (first paragraph), which suggests that the theory might only apply to external rotation. Somewhere there seems to be a contradiction. In short: What is the definition of a chiral active fluid as used e.g. in the title? Does it comprise both types of rotation?

This is a critical point, which we thank the reviewer for pointing out. We have only addressed the case in which total angular momentum is not conserved – we used to have this as an implicit assumption when considering “dry” active matter, but have now made it explicit in the manuscript text. The fluid can violate angular momentum conservation in one of two ways: as the reviewer points out by having an external field, such as a magnetic or an electric field. Alternatively, for a two-dimensional system near a solid substrate, angular momentum conservation could be broken due to strong rotational friction between the particles and the substrate. In that case, even if the rotation is “internal,” angular momentum conservation is violated and the hydrodynamic theory we develop also applies. As the reviewer correctly notes, if the particles are suspended at a two-dimensional fluid-fluid interface at which angular momentum is conserved, then the assumption of a “dry” active fluid that we use no longer holds. We hope that the new version of the manuscript makes more explicit the assumptions necessary to realize a chiral active fluid with odd viscosity.

2) As the authors discuss on page 2, their theory focuses on rotors with high spinning frequency and without a surrounding fluid (dry fluid). Translational friction is also ignored ($\Gamma^v = 0$, see end of first paragraph on page 2, top right). In this situation rotors quickly reach an equilibrium between applied torque and friction, and interaction of rotors are less important. It is my impression this excludes all systems but systems of driven grains in figure 1. Is this correct? Which experimental systems would the theory developed be applicable to?

This point is closely related to the previous one – we do only consider the case of a dry fluid, in which there may be two types of friction: rotational friction, which dissipates angular momentum, and translational friction, which dissipates linear momentum. The presence of odd viscosity is related to the dissipation of angular momentum only, and is independent of the translational friction. In this work, we have assumed translational friction to be small to simplify the analysis and emphasize the phenomena associated with odd viscosity only. However, we believe there are many active fluids in which the approximations that we have made do hold well. These may include both the macroscopic granular systems that the reviewer points out, as well as liquids composed of micro- and nano-scale particles near a substrate.

However, the value of odd viscosity that we derive, related to the angular momentum density, only applies in the case of granular fluids – the example that the reviewer mentions. For the examples of strongly interacting liquids, we believe odd viscosity could have a different value. Nevertheless, most of our work deals with the phenomenological consequences of odd viscosity, which would equally apply to spinning colloids, cells, or molecule, for example, if the system is near a momentum-dissipating substrate. We have modified the manuscript to state that the estimate for the value of odd viscosity in terms of the rotors’ parameters only applies to granular fluids, but that the phenomenology based on the hydrodynamic theory is more general.

3) Equation (5) of the manuscript appears identical to equation (41) of Avron (Ref. [13]). I thus wonder how much overlap of the current theory exists with the work in this reference? It is my impression the authors tackled a more general problem but I am a bit lost in the technical parts of the work presented in Supplementary Information, which covers more than 100 equations.

We agree with the referee that the equation that Eq. (5) in the manuscript is identical to Eq. (41) of Avron Ref. [13], both capturing the effects of odd viscosity on hydrodynamics. Thus, the novelty of our work is not in writing this equation down per se. Rather, as far as we are aware, we are the first ones to apply this equation to the hydrodynamics of active fluids. In our modifications of the manuscript, we have tried to emphasize this point, and the novelty of our work.

To summarize:

Previously, the one physical realization that has been proposed for an odd viscosity fluid is the two-dimensional fluid of electrons in an applied transverse magnetic field (i.e., quantum Hall fluid). In the context of active fluids, the novel features of odd viscosity are that it

(i) arises only out of equilibrium

(ii) is always accompanied by an antisymmetric stress

(iii) is not well defined as particles jam and active rotations are hindered by interactions.

Furthermore, as far as we are aware, we are the first to examine two of the phenomena in chiral active fluids in which we believe the effects of odd viscosity are likely to be observable – namely density variations in vortices, and the steady-state flow profile in shocks.

4) While the Supplementary Information is certainly helpful, I wonder if it is all needed? It appears to go significantly beyond the main text.

Although we agree with the reviewer that the Supplementary Information is rather technical, we hope that the new version of the manuscript helps distill some of the results of technical calculations in a way that is accessible without going through the derivations. In addition, we have moved the derivation of hydrodynamic equations from the SI to the Methods section of the main text.

5) The build-up of vortices has already been seen in experiment and simulation (e.g. Wensink et al. <https://journals.aps.org/pre/abstract/10.1103/PhysRevE.89.010302>; Refs [2,17,19]).

We thank the reviewer for this relevant reference, which we have now added.

6) Subfigures are not always discussed in the order of their appearance in the text. Some subfigures (2c, 2d) are not explicitly mentioned.

We have now corrected these mistakes.

7) At the beginning of the Supplementary Information covariant notation is sometimes used. However, this is not done in the main text or the later parts of the Supplementary Information. I think the convention should be kept consistent.

We agree – we have revised the SI in order to make the notation consistent throughout.

8) Would it be helpful to include the numerical solutions as a video instead of only the sections shown in the graphs of Figure 3?

We wholeheartedly agree that a movie for the evolution of the shock profile towards the steady state can help the reader better understand our simulations and results. We have now included a Supplementary Movie with our manuscript, which shows the evolution of shock profiles towards the steady states in Figs. 3a and 3c.

REVIEWERS' COMMENTS:

Reviewer #1 (Remarks to the Author):

In their resubmittal letter, the authors have responded in detail to the points raised in my previous report -- and I think to the points raised in the reports of the other referees as well. They have modified their manuscript accordingly, and substantially. This has led to a significant improvement of the manuscript. Therefore, I support publication of the manuscript in its present form.

Reviewer #3 (Remarks to the Author):

The authors have addressed all of my comments and concerns. I therefore recommend publication in Nature Communications without the need of further revisions.